# Causal Discovery in Linear Latent Variable Models Subject to Measurement Error

**Yuqin Yang**[*]
Georgia Institute of Technology

**AmirEmad Ghassami**
Johns Hopkins University

**Mohamed Nafea**
University of Detroit Mercy

**Negar Kiyavash**
École Polytechnique Fédérale
de Lausanne (EPFL)

**Kun Zhang**
Carnegie Mellon University
& MBZUAI

**Ilya Shpitser**
Johns Hopkins University

## Abstract

We focus on causal discovery in the presence of measurement error in linear systems where the mixing matrix, i.e., the matrix indicating the independent exogenous noise terms pertaining to the observed variables, is identified up to permutation and scaling of the columns. We demonstrate a somewhat surprising connection between this problem and causal discovery in the presence of unobserved parentless causes, in the sense that there is a mapping, given by the mixing matrix, between the underlying models to be inferred in these problems. Consequently, any identifiability result based on the mixing matrix for one model translates to an identifiability result for the other model. We characterize to what extent the causal models can be identified under a two-part faithfulness assumption. Under only the first part of the assumption (corresponding to the conventional definition of faithfulness), the structure can be learned up to the causal ordering among an ordered grouping of the variables but not all the edges across the groups can be identified. We further show that if both parts of the faithfulness assumption are imposed, the structure can be learned up to a more refined ordered grouping. As a result of this refinement, for the latent variable model with unobserved parentless causes, the structure can be identified. Based on our theoretical results, we propose causal structure learning methods for both models, and evaluate their performance on synthetic data.

## 1 Introduction

Learning causal structure among the variables of a system from observational data has received considerable attention in the literature, since subject matter knowledge on causal relationships is often incomplete or impossible to obtain in many applications [24, 16]. In many real-life systems, not all variables of interest or all direct common causes of those variables can be observed. This necessitates approaches for structure learning capable of dealing with latent variables. For the case that no a-priori restrictions on the functional form of causal mechanisms are imposed, constraint-based algorithms such as the Fast Causal Inference (FCI) algorithm have been proposed [24]. Such approaches are often unable to identify the direction of the majority of causal connections in the system, which motivates placing additional, often parametric, restrictions on the generating model. One of the most commonly used restrictions is to assume that the model is linear with non-Gaussian exogenous noise terms [21, 22, 12, 10]. This generating model leads to identification of the existence and orientation of all edges in a causal model, assuming that no latent variables that are common causes of two or more variables exist [21, 22]. Moreover, in such models even when latent variable common causes exist, it is still possible to orient additional edges compared to algorithms such as FCI [12, 19, 1].

---

[*]Correspondence to: yuqinyang@gatech.edu

36th Conference on Neural Information Processing Systems (NeurIPS 2022).

In certain applications, although some of the variables of interest (called underlying variables) are latent, we may have access to noisy measurements of them [17, 13, 29, 18, 11]. In this case, conditional independence patterns among observed variables (that are noisy measurements of underlying variables) are different from those among the underlying (measurement-error-free) variables. In general, constraint-based approaches to causal discovery are not able to correct for the difference in observed independence due to the presence of measurement error, and thus will produce erroneous edge adjacencies and orientations. Similarly, naive application of methods based on non-Gaussian exogenous noise assumption may also fail in recovering the correct causal relations in the presence of measurement error. This is due to the fact that the extra measurement noise terms break the asymmetry utilized by those methods.

In this paper, we bring additional insight into the problem of causal discovery under measurement error by showing a surprising connection between linear structural equation models (SEMs) where variables are measured with error, and linear SEMs with hidden variables. In particular, we consider two linear SEMs. In the first model, the unobserved variables are in fact of interest and can appear in any part of the causal structure of the system, but we assume that noisy measurements of these unobserved variables are available. We refer to this model as the linear SEM with measurement error (SEM-ME). The second model does not require measurements of the latent variables, as they are not of primary interest, yet we assume that all latent variables are root variables, meaning that they are not causally affected by any other latent or observed variables in the system. We refer to this model as the linear SEM with unobserved roots (SEM-UR). It is noteworthy that assuming latent variables are limited to be root variables does not affect the estimated total causal effects among the observed variables (see Remark 1 in Section 2.2), and provides a representation of a latent variable model that is consistent with these causal effects. For this reason, this restriction is common in the literature of causal inference [8, 9, 12, 5].

We study the identification of linear SEM-MEs and SEM-URs in a setup where the independent exogenous noise terms that causally (directly or indirectly) affect each observed variable can be identified. That is, the mixing matrix of the linear system that transforms exogenous noise terms to observed variables is identified up to permutation and scaling of the columns. This can be satisfied, for example, if all independent exogenous noise terms are non-Gaussian. Our first main contribution is presenting a mapping between the linear SEM-MEs and SEM-URs, which demonstrates a correspondence between a weighted causal diagram[2] generated by SEM-UR and a set of weighted causal diagrams from the SEM-ME (Theorem 1). The models in this correspondence all possess the same mixing matrix. Consequently, any identifiability result based on the mixing matrix for one model is applicable to the other model. This allows us to study the problem of causal discovery in these two models together.

Additionally, we study the identifiability of linear SEM-MEs and SEM-URs, and show how it benefits from a two-part faithfulness assumption. The first part prevents existence of zero total causal effects of a variable on its descendants and the second part prevents cancellation or proportionality among specific edges. Our second main contribution is to characterize the extent of identifiability of the causal model under our faithfulness assumption. We demonstrate that if only the first part of the faithfulness assumption is imposed, the model can be learned up to an equivalence class characterized by an ordered grouping of the variables which we call *ancestral ordered grouping* (AOG). In this grouping, the induced graph on the variables of each group is a star graph (see Section 4.3). Although the AOG is identified, not all edges across the groups or the group centers (or their exogenous noise terms) are identifiable (Theorem 2). The AOG characterization is a refined version of the ordered grouping proposed in [29]. We further show that if both parts of the faithfulness assumption are imposed, the model can be learned up to an equivalence class characterized by a more refined ordered grouping which we call *direct ordered grouping* (DOG). Edges across groups can be identified under this characterization, yet the the group centers (or their exogenous noise terms) remain unidentified (Theorem 3). This characterization further implies that the ground-truth structure of a SEM-UR is uniquely identifiable (Corollary 3). Lastly, we show that models in the DOG equivalence class are strictly sparser than other models in the AOG equivalence class (Proposition 4). We propose causal structure learning methods for both SEM-ME and SEM-UR based on this property, and evaluate their performance on synthetic data.

---

[2] A weighted causal diagram is the directed graph corresponding to the linear model which associates the coefficient of variable $X_1$ in the structural equation for $X_2$ to the edge from $X_1$ to $X_2$.

## 2 Model description

In this section, we first introduce the two models considered in this paper, namely linear SEM-ME and SEM-UR. We then discuss a model assumption needed for our identifiability results.

### 2.1 Linear structural equation model with measurement error (SEM-ME)

**Definition 1 (Linear SEM-ME)** *A linear SEM-ME consists of a set of "underlying" variables which can be partitioned into unobserved underlying variables $\mathcal{Z} = \{Z_1, ..., Z_{p_z}\}$ and observed underlying variables $\mathcal{Y} = \{Y_1, ..., Y_{p-p_z}\}$. In addition, we have another set of observed variables $\mathcal{U} = \{U_1, \cdots, U_{p_z}\}$ corresponding to the noisy measurements of $\mathcal{Z}$. The underlying variables in $\mathcal{Z} \cup \mathcal{Y}$ can be arranged in a total (causal) order (such that no later variable in the order can cause any earlier variable), and variables $V_i \in \mathcal{Z} \cup \mathcal{Y}$ and $U_i \in \mathcal{U}$ are generated as follows:*

$$V_i = \sum_{j:\, V_j \in Pa(V_i)} c_{ij} V_j + N_{V_i} \quad i \in [p]; \qquad U_i = Z_i + N_{U_i}, \quad i \in [p_z], \tag{1}$$

*where $[n] := \{1, 2, \cdots, n\}$, $Pa(V_i) \subseteq \mathcal{Z} \cup \mathcal{Y}$ denotes direct causes of $V_i$. $N_{V_i}$ (resp. $N_{U_i}$) is the exogenous noise term (resp. the measurement error) corresponding to $V_i$ (resp. $U_i$).*

We define the weighted causal diagram of the linear SEM-ME as a weighted directed graph where the nodes are the variables in $\mathcal{Z} \cup \mathcal{Y}$. There is a directed edge (causal connection) from $V_i$ to $V_j$ with weight $c_{ji}$ if and only if $c_{ji} \neq 0$, for $V_i, V_j \in \mathcal{Z} \cup \mathcal{Y}$. Because of our causal order assumption, the causal diagram will be acyclic. We use the terms node and variable interchangeably. The linear SEM-ME in this work is a generalization of the linear causal model with measurement error proposed in [29], in the sense that some of the underlying variables can be observed.

We define an *unobserved leaf node* (`u-leaf` node) as an unobserved underlying variable in $\mathcal{Z}$ that does not have any children in $\mathcal{Z} \cup \mathcal{Y}$. The rest of the underlying variables are referred to as *non u-leaf nodes* (`nu-leaf` nodes): These include unobserved non-leaf underlying variables, which have children in $\mathcal{Z} \cup \mathcal{Y}$, and observed underlying variables in $\mathcal{Y}$. Similar to the argument in [29], given observed data generated by the linear SEM-ME in Equation (1), for a `u-leaf` node $Z_i$, its exogenous noise term $N_{Z_i}$ is not distinguishable from its measurement error $N_{U_i}$. This follows because $Z_i$ is not observed, and $N_{Z_i}$ only influences one observed variable which is the noisy measurement $U_i$. Therefore, we restrict our focus to the following canonical form.

**Definition 2 (Canonical form of a linear SEM-ME)** *The canonical form of a linear SEM-ME is the one in which all `u-leaf` nodes have no exogenous noise terms. Specifically,*

$$\begin{bmatrix} Z^L \\ Z^{NL} \\ Y \end{bmatrix} = \begin{bmatrix} \mathbf{D} \\ \mathbf{C}_Z \\ \mathbf{C}_Y \end{bmatrix} Z^{NU} + \begin{bmatrix} 0 \\ N_{Z^{NL}} \\ N_Y \end{bmatrix}, \qquad U = \begin{bmatrix} Z^L \\ Z^{NL} \end{bmatrix} + \begin{bmatrix} N_{Z^L} + N_{U^L} \\ N_{U^{NL}} \end{bmatrix}, \tag{2}$$

*where $Z^{NL}$, $Z^L$, $Y$ are the vectors of unobserved underlying non-leaf variables, `u-leaf` nodes, and observed underlying variables, respectively. $N_{Z^{NL}}$, $N_{Z^L}$, and $N_Y$ are the corresponding noise vectors. $Z^{NU} = [Z^{NL}; Y]$ is the vector of `nu-leaf` nodes. $U$ is the vector of noisy measurements and $N_{U^{NL}}$ and $N_{U^L}$ denote the noise vectors corresponding to the measurements of $Z^{NL}$ and $Z^L$, respectively. $\mathbf{C}$ represents the causal connections among `nu-leaf` nodes, and can be partitioned into $[\mathbf{C}_Z; \mathbf{C}_Y]$ corresponding to $[Z^{NL}; Y]$, respectively. $\mathbf{D}$ represents the causal connections from `nu-leaf` nodes to `u-leaf` nodes. Note that $\mathbf{C}$ can be arranged into a strictly lower triangular matrix via simultaneous column and row permutations (due to the acyclicity assumption).*

### 2.2 Linear structural equation model with unobserved roots (SEM-UR)

**Definition 3 (Linear SEM-UR)** *A linear SEM-UR consists of a set of observed variables $\mathcal{X} = \{X_1, \cdots, X_q\}$ and another set of latent variables $\mathcal{H} = \{H_1, \cdots, H_l\}$. All latent variables are assumed to be root variables (have no direct causes). The observed variables are arranged in a causal order, and each observed variable is directly influenced by a linear combination of other observed and latent variables, plus an independent exogenous noise term.*

$$H_i = N_{H_i}, \quad i \in [l]; \qquad X_j = \sum_{i \in [l]} b_{ji} H_i + \sum_{k < j} a_{jk} X_k + N_{X_j}, \qquad j \in [q]. \tag{3}$$

We define the weighted causal diagram of the linear SEM-UR as a directed graph where the nodes are the observed and latent variables in $\mathcal{X} \cup \mathcal{H}$. There is a causal connection from $X_i$ (resp. $H_i$) to $X_j$ with weight $a_{ji}$ (resp. $b_{ji}$) if and only if $a_{ji}$ (resp. $b_{ji}$) $\neq 0$. Because of our causal order assumption, the causal diagram will be acyclic. Note that unlike in [12] and [19], we consider a broader model

which allows latent variables to have a single observed child. However, unlike the works considering relations among latent variables [1, 26, 3, 27], we restrict all latent variables to roots; see Remark 1.

A linear SEM-UR can be written in the following matrix form:

$$H = N_H; \qquad X = \mathbf{B}H + \mathbf{A}X + N_X, \tag{4}$$

where $X = [X_1 \ \cdots \ X_q]^\top$ and $H = [H_1 \ \cdots \ H_l]^\top$ are the vectors of observed and latent variables, respectively. $N_X = [N_{X_1} \ \cdots \ N_{X_q}]^\top$ and $N_H = [N_{H_1} \ \cdots \ N_{H_l}]^\top$ represent the vectors of independent noise terms associated with $X$ and $H$, respectively. $\mathbf{B}$ represents the causal connections from latent to observed variables, and $\mathbf{A}$ represents the causal connections among observed variables, which can be assumed to be a strictly lower-triangular matrix (due to the acyclicity assumption).

**Remark 1** *There exist works aiming to find the causal relations among the latent variables (as well as their relations with the measured variables) [23, 26, 3, 1, 27]. Our definition of a linear SEM-UR requires all the latent variables to be parentless. This assumption is common in the literature of causal inference, as it does not restrict the feasible total causal effects (i.e., sum of products of path coefficients) among the observed variables [8, 9, 5]. Specifically, let us start from a general linear latent variable model in which some latent variables have parents. Hoyer et al. [12] proposed an algorithm which maps such general model to one in which latent variables are parentless. They showed that the resulting model is observationally and causally equivalent to the original model, i.e., the joint distributions of observed variables are identical, and all causal effects of observed variables on other observed variables are identical in the models before and after the mapping.*

### 2.3 Separability assumption

For both models in Sections 2.1 and 2.2, we assume that the noise components can be separated from observations (i.e., the mixing matrices are recoverable). We first describe the mixing process for each of the two models, and then state our separability assumption.

From Equation (2), we can write all underlying variables in the linear SEM-ME in terms of the independent noise terms, as follows.

$$\begin{bmatrix} Z^L \\ Z^{NL} \\ Y \end{bmatrix} = \mathbf{W}^{ME} \begin{bmatrix} N_{Z^{NL}} \\ N_Y \end{bmatrix}, \qquad \text{where} \quad \mathbf{W}^{ME} = \begin{bmatrix} \mathbf{D}(\mathbf{I} - \mathbf{C})^{-1} \\ (\mathbf{I} - \mathbf{C})_Z^{-1} \\ (\mathbf{I} - \mathbf{C})_Y^{-1} \end{bmatrix}. \tag{5}$$

$\mathbf{I}$ is the identity matrix, and $(\mathbf{I} - \mathbf{C})_Z^{-1}$ and $(\mathbf{I} - \mathbf{C})_Y^{-1}$ represent the rows of $(\mathbf{I} - \mathbf{C})^{-1}$ corresponding to $Z^{NL}$ and $Y$, respectively. Using the relation between $U$ and $Z$ in Equation (2), the two types of observed variables, i.e., measurements $U$ and observed underlying variables $Y$, can be written as

$$\begin{bmatrix} U \\ Y \end{bmatrix} = \underbrace{\begin{bmatrix} \mathbf{W}^{ME} & \mathbf{I} \\ & \mathbf{0} \end{bmatrix}}_{\mathbf{W}} \begin{bmatrix} N_{Z^{NL}} \\ N_Y \\ N_{Z^L} + N_{U^L} \\ N_{U^{NL}} \end{bmatrix}. \tag{6}$$

We refer to $\mathbf{W}$ as the overall mixing matrix of the system. Given any column permuted and rescaled version of $\mathbf{W}$, we can recover matrix $\mathbf{W}^{ME}$ by removing the submatrix $[\mathbf{I} \ \ \mathbf{0}]^\top$ of size $p \times p_z$ corresponding to $U$. Recall that $p_z$ is the cardinality of $Z$ (which is the same as the cardinality of $U$).

Similarly, according to Equation (4), observed variables in the linear SEM-UR can be written in terms of the independent noise terms as

$$X = \mathbf{W}^{UR} \begin{bmatrix} N_H \\ N_X \end{bmatrix}, \qquad \text{where} \quad \mathbf{W}^{UR} = \left[ (\mathbf{I} - \mathbf{A})^{-1}\mathbf{B} \quad (\mathbf{I} - \mathbf{A})^{-1} \right]. \tag{7}$$

**Assumption 1 (Separability)** *The linear SEM-ME (resp. linear SEM-UR) is separable, that is, the mixing matrix $\mathbf{W}^{ME}$ in Equation (5) (resp. $\mathbf{W}^{UR}$ in Equation (7)) can be recovered from observations of $[U; \ Y]$ (resp. $X$) up to permutation and scaling of its columns.*

Separability assumption states that for every observed mixture, the independent exogenous noise terms pertaining to this mixture can be separated, i.e., the mixing matrix can be recovered up to permutation and scaling of its columns. An example of a setting where this assumption holds is when all exogenous noises are non-Gaussian. In this case, if the model satisfies the requirement in [7, Theorem 1] (SEM-ME always does), overcomplete Independent Component Analysis (ICA) can be used to recover the mixing matrix up to permutation and scaling of its columns. Another

example where separability assumption is satisfied is the setup in which the noise terms are piecewise constant functionals satisfying a set of mild conditions [2]. On the other hand, an example where this assumption is violated is when all exogenous noise terms have Gaussian distributions. In this case, the mixing matrix can only be recovered up to an orthogonal transformation.

## 3 Mapping between the weighted graphs of the models for identifiability

A key observation in this work is that there exists a mapping between the weighted causal diagrams of the introduced SEM-ME and SEM-UR, which leads to the corresponding mixing matrices $\mathbf{W}^{ME}$ and $\mathbf{W}^{UR}$ being transposes of one another.

Define the mapping $\varphi$ from the set of weighted causal diagrams of linear SEM-URs, denoted by $\mathcal{M}^{UR}$, to the set of weighted causal diagrams of canonical linear SEM-MEs, denoted by $\mathcal{M}^{ME}$, where for each diagram $M \in \mathcal{M}^{UR}$, $\varphi(M) \subset \mathcal{M}^{ME}$ is constructed as follows: (a) Replace each latent variable in $M$ with a $Z$ variable. (b) Replace each observed root variable in $M$ with a $Y$ variable. (c) Replace the rest of the observed variables in $M$ with either $Z$ or $Y$ variables. (d) Reverse all the edges in $M$. Note that $\varphi(M)$ encompasses a set of models, corresponding to the possible choices in step (c).

Similarly define the mapping $\phi$ from the set $\mathcal{M}^{ME}$ to the set $\mathcal{M}^{UR}$, where for each diagram $M' \in \mathcal{M}^{ME}$, $\phi(M') \in \mathcal{M}^{UR}$ is constructed as follows. (a) Replace each u-leaf variable in $M'$ with a latent variable. (b) Replace the rest of the variables in $M'$ with observed variables. (c) Reverse all the edges in $M'$. Note that from the definition of the two mappings, for any given $M \in \mathcal{M}^{UR}$, $M = \phi(\varphi(M))$. For any given $M' \in \mathcal{M}^{ME}$, $M' \in \varphi(\phi(M'))$.

**Theorem 1** *Let $M \in \mathcal{M}^{UR}$ with mixing matrix $\mathbf{W}^{UR}(M)$ and $M'$ be a corresponding element in $\mathcal{M}^{ME}$ with mixing matrix $\mathbf{W}^{ME}(M')$, where $M' \in \varphi(M)$ and $M = \phi(M')$. Then there exist a permutation of the columns of $\mathbf{W}^{UR}(M)$, denoted by $\tilde{\mathbf{W}}^{UR}(M)$, and a permutation of the columns of $\mathbf{W}^{ME}(M')$, denoted by $\tilde{\mathbf{W}}^{ME}(M')$, which satisfy $\tilde{\mathbf{W}}^{ME}(M') = (\tilde{\mathbf{W}}^{UR}(M))^\top$.*

The proofs of all the results are provided in the Appendix. Note that as it can be seen from the proof of Theorem 1, whether the nu-leaf variables, that are non-leaf in the SEM-ME, are observed or not does not change the corresponding mixing matrix.

**Remark 2** *The mapping introduced here is between a weighted diagram in SEM-UR and a set of weighted diagrams in SEM-ME. The elements of the set in fact correspond to different settings of the non-leaf underlying variables being observed or not. Note that in most applications, it is clear to the researcher whether an underlying variable is observed or measured with error. Therefore, only one element of the set is relevant to the application and the set reduces to a single element. Consequently, in our identifiability results, without loss of generality, we assume the set is a singleton.*

Theorem 1 implies that under separability assumption, any identifiability result based on the mixing matrix for one model is applicable to the other model, by reversing all edges. This allows us to study the problem of causal discovery of these two models simultaneously.

**Example 1** *Consider the linear SEM-UR in Figure 1(a) comprised of three observed and one latent variable (denoted by a single circle). The corresponding linear SEM-ME under the mapping $\varphi$ is shown in Figure 1(b). Specifically, (i) the latent variable $H$ is mapped to the u-leaf variable $Z_L$ (denoted by a double circle); (ii) the observed root $X_1$ is mapped to the observed underlying variable $Y_1$; and (iii) observed variables $X_2$ and $X_3$ could be mapped to either observed or unobserved underlying variables; hence the UR graph will be mapped to a set of 4 graphs in the ME model. The mixing matrices for both models are shown in Figure 1. It can be readily seen that there exists a permutation of the columns of $\mathbf{W}^{UR}$, which is equal to the transpose of $\mathbf{W}^{ME}$.*

$$\mathbf{W}^{UR} = \begin{bmatrix} 0 & 1 & 0 & 0 \\ d & a & 1 & 0 \\ bd+c & ab & b & 1 \end{bmatrix}$$

(a) Linear SEM-UR

$$\mathbf{W}^{ME} = \begin{bmatrix} 0 & d & bd+c \\ 0 & 1 & b \\ 0 & 0 & 1 \\ 1 & a & ab \end{bmatrix}$$

(b) Linear SEM-ME

Figure 1: Example explaining the mapping.

## 4 Identifiability of models

In this section, we study the extent of identifiability of linear SEM-ME and SEM-UR under the separability assumption. We consider both a weaker notion of structure identification, where only

the graph structure of the model is obtained, and a stronger notion of model identification where, in addition to the graph structure, model parameters are recovered as well. In Section 4.1, we propose SEM-ME and SEM-UR faithfulness assumptions, which consist of two parts, the first of which is identical to the conventional faithfulness in linear causal models. We first discuss model identifiability under conventional faithfulness (i.e., the first part of our assumptions), which can be characterized by *ancestral ordered grouping* (AOG) of the variables, explained in Section 4.2. Next, under both parts of our proposed faithfulness assumptions, the extent of identifiability can be characterized by *direct ordered grouping* (DOG) of the variables, explained in Section 4.3. This leads to a subsequent result on structure identifiability for SEM-UR. In Section 4.4, we show that the DOG characterization possesses a sparsity property, which can be used to construct recovery algorithms for both models.

## 4.1 Faithfulness assumption

**Assumption 2 (SEM-ME faithfulness)** *(a) The total causal effect of any underlying variable on its descendant is not equal to zero. (b) For each variable $V$ in a SEM-ME, the causal effect of parents of $V$ on $V$ cannot be replicated by fewer or equal variables due to fine tuning of the parameters. (See Appendix B.1 for the formal statement.)*

**Assumption 3 (SEM-UR faithfulness)** *(a) The total causal effect of any observed or latent variable on its descendant is not equal to zero. (b) For each variable $V$ in a SEM-UR, the causal effect of $V$ on children of $V$ cannot be replicated by fewer or equal variables due to fine tuning of the parameters. (See Appendix B.1 for the formal statement.)*

Assumptions 2*(a)* and 3*(a)* are identical to conventional faithfulness in linear causal models. They require that when multiple causal paths exist from any (observed or unobserved) variable to its descendants, their combined effect (i.e., sum of products of path coefficients) is not equal to zero, see [24]. Assumptions 2*(b)* and 3*(b)* prevent certain path cancellation or parameter proportionality. Importantly, Assumptions 2 and 3 are violated with probability zero if all model coefficients are drawn randomly and independently from continuous distributions; see Appendix B.3 for the proof. As an example for violation of Assumption 2*(b)*, consider the linear SEM-ME generated by the graph in Figure 2(a), which satisfies Assumption 2*(a)*. However, we can write $Z_4$ as $(b + c)Z_2 + N_{Z_3}$, which is caused by the path cancellation of the triangle $(Z_1, Z_3, Z_4)$. This means that that the causal effect of $Z_1$ and $Z_2$ on $Z_4$ can be summarized by $Z_2$ alone. Therefore, the model violates Assumption 2*(b)*. Lastly, Assumption 3 is strictly weaker than bottleneck faithfulness in [1]; see Appendix B.4 for the proof. Please refer to Appendix B for more discussion about our faithfulness assumption and Appendix D for a detailed comparison between our identifiability results and the results in [1].

## 4.2 Ancestral ordered grouping

We first study the extent of identifiability of the models only under conventional faithfulness, i.e., Assumptions 2*(a)* and 3*(a)*.

**Definition 4 (Ancestral ordered grouping (AOG))** *The AOG of a SEM-ME (resp. SEM-UR) is a partition of $\mathcal{Z} \cup \mathcal{Y}$ (resp. $\mathcal{H} \cup \mathcal{X}$) into distinct sets. This partition is described as follows: (1) Assign each* `nu-leaf` *node (resp. observed node) to a distinct group. (2) SEM-ME: For each* `u-leaf` *node $Z_j \in \mathcal{Z}$, if there exists one parent $V_i$ such that $Z_j$ has no other parents, or all other parents of $Z_j$ are also ancestors of $V_i$, assign $Z_j$ to the same group as $V_i$. Otherwise, assign $Z_j$ to a separate group (with no* `nu-leaf` *node). (2) SEM-UR: For each*

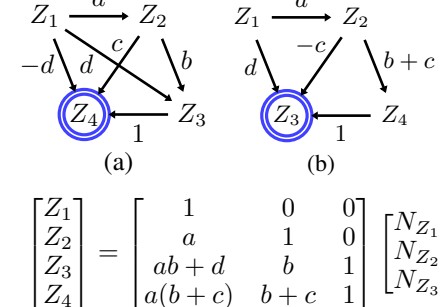

$$\begin{bmatrix} Z_1 \\ Z_2 \\ Z_3 \\ Z_4 \end{bmatrix} = \begin{bmatrix} 1 & 0 & 0 \\ a & 1 & 0 \\ ab+d & b & 1 \\ a(b+c) & b+c & 1 \end{bmatrix} \begin{bmatrix} N_{Z_1} \\ N_{Z_2} \\ N_{Z_3} \end{bmatrix}$$

Figure 2: Example explaining AOG and SEM-ME faithfulness assumption.

*latent variable $H_j$, if there exists one child $X_i$ such that $H_j$ has no other children, or all other children of $H_j$ are also descendants of $X_i$, assign $H_j$ to the same group as $X_i$. Otherwise, assign $H_j$ to a separate group (with no observed variable).*

**Definition 5 (AOG equivalence class)** *The AOG equivalence class of a linear SEM-ME (resp. SEM-UR) is a set of models where the elements of this set all have the same mixing matrix (up to permutation and scaling) and same ancestral ordered groups.*

For example, the two models represented by the graphs in Figure 2 have the same mixing matrix and ancestral ordered groups. The elements of an AOG equivalence class possess the following property.

**Proposition 1** *Models in the same AOG equivalence class have the same causal order among the groups, but not necessarily all the same edges across the groups.*

The AOG characterization is a refined version of the original ordered grouping proposed in [29], where no variables in later groups cause variables in earlier groups. Yet, the grouping introduced in that work is order dependent and hence in general not unique. Moreover, it does not always characterize the extent of identifiability. Our AOG characterization also uses a similar idea as the learning approach in [19]; however, that work allows for latent variables which have parents and in general does not recover the full structure. Note that the AOG can be recovered based on the support of the mixing matrix under Assumption 2*(a)* (or 3*(a)*). Please refer to Appendix A for more details.

According to Definition 4, there is at most one `nu-leaf` node in each ancestral ordered group of a SEM-ME. Furthermore, each `u-leaf` node $Z_j$ is assigned to the ancestral ordered group of at most one of its parents: Following the true causal order among `nu-leaf` nodes, only the last parent of $Z_j$ may have the same ancestor set. Hence, if a group has more than one node, then there must be exactly one `nu-leaf` node, and the rest of the nodes are `u-leaf` nodes which are children of this node. This concludes that the induced structure on each ancestral ordered group is a star graph. Similar property holds for SEM-UR. Define the center of the ancestral ordered group as the `nu-leaf` node (resp. the observed node), or the `u-leaf` node (resp. latent node) if the group is of size one. The following result illustrates that fixing the center of the ancestral ordered groups for SEM-ME, and fixing the exogenous noise term of the center of the ancestral ordered groups as well as the choice of scaling and permutation of the columns of $\mathbf{B}$ for SEM-UR, leads to unique identification of the models.

**Proposition 2** *(i) Models in an AOG equivalence class of a SEM-ME can be identified by the choice of the centers of the groups. That is, for a given choice of the centers, there is only one corresponding model. (ii) Models in an AOG equivalence class of a SEM-UR can be identified (up to the permutation and scaling of the columns of $\mathbf{B}$ in* (4)*) by the choice of the exogenous noise terms associated to the centers of the groups. That is, for a given assignment of the exogenous noise terms, all corresponding models have the same $\mathbf{A}$ in* (4)*, and have the same $\mathbf{B}$ up to permutation and scaling of the columns.*

Equipped with Proposition 2, we are now ready to state our identifiability results for SEM-ME and SEM-UR under Assumptions 2*(a)* and 3*(a)*), respectively.

**Theorem 2** *Under Assumptions 1 and 2(a) (resp. 3(a)), the linear SEM-ME (resp. SEM-UR) can be identified up to its AOG equivalence class.*

**Corollary 1** *(i) Denote the AOG of a linear SEM-ME (resp. SEM-UR) as $\{\mathcal{G}^{(g)}\}_{g \in [g_k]}$, where $g_k$ is the number of ancestral ordered groups. The size of the AOG equivalence class, described in Theorem 2, is equal to $\prod_{g \in [g_k]} |\mathcal{G}^{(g)}|$. (ii) Under Assumptions 1 and 2(a), a linear SEM-ME can be uniquely identified if and only if no `u-leaf` node has precisely the same ancestors as any `nu-leaf` node. (iii) Under Assumptions 1 and 3(a), a linear SEM-UR can be identified up to the permutation and scaling of the columns of $\mathbf{B}$ if and only if no latent variable has precisely the same descendants as any observed variable.*

As stated in Theorem 2, in general, the edges across the groups cannot all be discovered and the centers of the groups (or their corresponding exogenous noise terms) are not identifiable. This is due to the fact that when only Assumption 2*(a)* holds, certain path cancellation or parameter proportionality in the model can occur. This is illustrated in the following example.

**Example 2** *Consider the SEM-ME generated by causal graph shown in Figure 2(a). The AOG is $\{\{Z_1\}, \{Z_2\}, \{Z_3, Z_4\}\}$. Because either $Z_3$ or $Z_4$ can be the center of the last group, the AOG equivalence class includes the ground-truth and the model represented by Figure 2(b). Since both models have the same mixing matrix, an identification algorithm merely based on the mixing matrix and Assumption 2(a) cannot distinguish these two models. Note that not only can we not learn the direction of the edge in group $\{Z_3, Z_4\}$, but also the existence of some of the edge across groups, such as $Z_1 \to Z_4$, cannot be established in the ground-truth because Assumption 2(b) is violated.*

### 4.3 Identifiability of the model up to equivalence classes

Under Assumptions 1 and 2 (resp. 3), the extent of identifiability of a SEM-ME (resp. SEM-UR) can be characterized by the direct ordered grouping defined in Definition 6 below. We first present two conditions that are used to implement the DOG. We then give the formal definition of DOG. In the following, we use $Ch(V)$ to denote the set of children of variable $V$ in the causal diagram.

**Condition 1 (SEM-ME edge identifiability)** *For a given edge from an `nu-leaf` node $V_i$ to a `u-leaf` node $Z_l$, at least one of the following two conditions is satisfied: (a) $Pa(Z_l) \setminus \{V_i\}$ is not a subset of $Pa(V_i)$. That is, there exists another parent $V_j$ of $Z_l$, which is not a parent of $V_i$. (b) $Pa(Z_l)$ is not a subset of $\cap_{V_k \in Ch(V_i) \setminus \{Z_l\}} Pa(V_k)$. That is, there exists a child $V_k$ of $V_i$ and a parent $V_j$ of $Z_l$ such that $V_j$ is not a parent of $V_k$.*

**Condition 2 (SEM-UR edge identifiability)** *For a given edge from a latent variable $H_l$ to an observed variable $X_i$, at least one of the following two conditions is satisfied: (a) $Ch(H_l) \setminus \{X_i\}$ is not a subset of $Ch(X_i)$. That is, there exists another observed child $X_j$ of $H_l$, which is not a child of $X_i$. (b) $Ch(H_l)$ is not a subset of $\cap_{X_k \in Pa(X_i) \setminus \{H_l\}} Ch(X_k)$. That is, there exists an observed (or latent) parent $X_k$ (or $H_k$) of $X_i$ and a child $X_j$ of $H_l$ such that $X_j$ is not a child of $X_k$ (or $H_k$).*

Figure 3(b) demonstrates an example of a graph structure which satisfies Assumption 2 while containing an edge which violates Condition 1. Condition 1 is an equivalent formulation of the conditions for unique identifiability derived in [29]. In that work, Zhang et al. proved that a linear SEM-ME can be uniquely recovered if Condition 1 is satisfied for every edge from an `nu-leaf` node to a `u-leaf` node.[3] They also conjectured that this condition is necessary for identifiability. We prove their conjecture here under a slightly different faithfulness assumption. Furthermore, we demonstrate that the conditions can be used to characterize an equivalence class of the models which characterizes the extent of identifiability under Assumptions 1 and 2; the same assertion holds for linear SEM-UR. This characterization is done based on the notion of direct ordered grouping, defined as follows.

**Definition 6 (Direct ordered grouping (DOG))** *The DOG of a linear SEM-ME (resp. SEM-UR) is a partition of $\mathcal{Z} \cup \mathcal{Y}$ (resp. $\mathcal{H} \cup \mathcal{X}$) into distinct sets. This partition is described as follows:*
*(1) Assign each `nu-leaf` node (resp. observed node) to a distinct ordered group.*
*(2) SEM-ME: For each `u-leaf` node $Z_j \in \mathcal{Z}$, if there exists one parent $V_i$ such that the edge from $V_i$ to $Z_j$ violates Condition 1, assign $Z_j$ to the same ordered group as $V_i$. Otherwise, assign $Z_j$ to a separate ordered group (with no `nu-leaf` node).*
*(2) SEM-UR: For each latent variable $H_j$, if there exists one child $X_i$ such that the edge from $H_j$ to $X_i$ violates Condition 2, assign $H_j$ to the same ordered group as $X_i$. Otherwise, assign $H_j$ to a separate ordered group (with no observed variable).*

**Definition 7 (DOG equivalence class)** *The DOG equivalence class of a linear SEM-ME (resp. SEM-UR) is a set of models where the elements of this set all have the same mixing matrix (up to permutation and scaling) and same direct ordered groups.*

For example, the two models represented by the graphs in Figure 2 have the same mixing matrix, but different direct ordered groups. We have the following property regarding elements of an DOG equivalence class.

**Proposition 3** *Models in the same DOG equivalence class have the same edges across the groups.*

The counterpart of Proposition 2 holds for the case of direct ordered groups as well and we avoid repeating it. Based on that counterpart, we have the following result regarding identifiability of linear SEM-ME and SEM-UR under Assumptions 2 and 3, respectively.

**Theorem 3** *Under Assumptions 1 and 2 (resp. 3), the linear SEM-ME (resp. SEM-UR) can be identified up to its DOG equivalence class.*

Theorem 3 states that we can learn the structure among the groups, but the center of the group (or the noise term corresponding to it) will remain unidentified. Unlike the case of the AOG equivalence class, all the edges across the groups will be identified. That is, the choice of center does not change the edges across groups in DOG equivalence class, while it may change the edges in AOG equivalence class. Further, the sizes of the direct ordered groups are smaller than the sizes of the ancestral ordered groups. As an example, for the model shown in Figure 2(b), all direct ordered groups are of size one.

**Corollary 2** *(i) The counterpart of Corollary 1(i) is true for the case of DOG as well and we do not repeat it here. (ii) Under Assumptions 1 and 2, a SEM-ME can be uniquely identified if and only if for every edge from an `nu-leaf` node to a `u-leaf` node, Condition 1 is satisfied. (iii) Under Assumptions 1 and 3, a SEM-UR can be identified up to the permutation and scaling of the columns of $\mathbf{B}$ if and only if for every edge from a latent variable to an observed variable, Condition 2 is satisfied.*

**Identifiability of the structure of the SEM-UR.** As shown in Theorem 3, for a SEM-UR, the only undetermined part in the DOG equivalence class pertains to the assignment of the exogenous noises and coefficients, but the structure is the same. Consequently, if only the identification of the structure without weights is of interest, Assumptions 1 and 3 are sufficient. See Figure 3(a) for an example of two DOG equivalent SEM-URs. This identifiability result is summarized in the following corollary.

**Corollary 3** *Under Assumptions 1 and 3, the structure of a SEM-UR can be uniquely recovered.*

---

[3]Note that in their model, Zhang et al. assumed that all the underlying variables are unobserved.

### 4.4 Recovery algorithm

We note that our faithfulness assumption implies that the ground-truth model has strictly fewer edges than any model that has the same mixing matrix and satisfies conventional faithfulness (i.e., in the AOG equivalence class) but does not belong to the DOG equivalence class. This property can be leveraged to recover the ground-truth model or a member of its DOG equivalence class.

**Proposition 4** *Suppose a SEM-ME (resp. SEM-UR) satisfies Assumptions 1 and 2 (resp. 3). Any model that belongs to the same AOG equivalence class but does not belong to the same DOG equivalence class has strictly more edges than any member in the DOG equivalence class.*

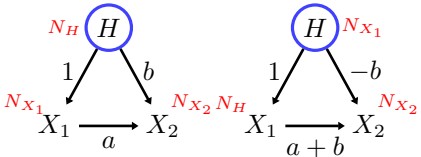

(a) Equivalence class of a SEM-UR

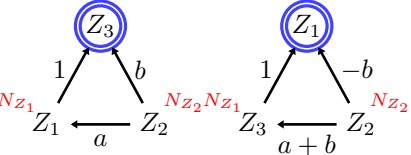

(b) Equivalence class of a SEM-ME

Figure 3: DOG Equivalence classes.

The steps of our recovery algorithm are as follows: (1) Recover the mixing matrix $\mathbf{W}^{ME}$ (resp. $\mathbf{W}^{UR}$) from observational data. (2) Return the AOG of the true model by comparing the support across different rows/columns in $\mathbf{W}^{ME}$ (resp. $\mathbf{W}^{UR}$). See Appendix A.2 for details and pseudo-code for AOG recovery. (3) For all possible choices of the center (resp. noise term associated to the center) of each ancestral ordered group, find a choice that leads to the graph with fewest number of edges in the recovery output (see the proof of Lemma 1).

## 5 Simulations

We evaluated the performance of our recovery algorithm on randomly generated linear SEM-MEs and SEM-URs with different number of variables.[4] We considered two cases: (1) when a noisy version of the mixing matrix is given, where the noise is Gaussian with different choices of variance denoted by $d^2$, and (2) when synthetic data comes from a linear generating model with non-Gaussian noises with different sample sizes, and the mixing matrix needs to be estimated. We used the Reconstruction ICA algorithm [14] as our overcomplete ICA method. For Case (1), we compared our recovery method (which we refer to as DOG) with an approach based on AOG equivalence class (similar to the learning method in [19]). In addition to the AOG method, for Case (2), we compared our method with ICA-LiNGAM [21]. Note that the ICA-LiNGAM method is not designed for systems with latent variables, but benefits from strong performance for recovering the mixing matrix in causally sufficient settings. The goal of this comparison was to demonstrate the necessity of using methods designed specifically to handle latent variables.

We compared the recovery of the adjacency matrix to the ground-truth, where we used normalized structural Hamming distance (SHD/Edge) and F1 score as our performance metrics. Our results for Cases (1) and (2) are shown in top and bottom two rows in Figure 4, respectively. As seen in these figures, our recovery algorithm outperforms recovery algorithm based on AOG. In particular, our method recovers the structure in SEM-UR model, and can recover part of the structure in SEM-ME that is shared among DOG equivalence classes. Moreover, both methods outperform ICA-LiNGAM. Please refer to Appendix E for additional results and analysis.

## 6 Conclusion and Discussion

In real-world applications, we do not observe the exact value of all relevant variables; the measurements of some variables are prone to errors, or some other variables cannot be observed altogether. For example, in neuroscience and genomics, measured brain signals obtained by functional magnetic resonance imaging (fMRI) or the measured gene expression using RNA sequencing usually contain errors through the measurement process [18, 29]. Other examples include responses to psychometric questionnaires where the questions represent noisy views of various traits [1, 20], and returns in stock market, where they may be confounded by several unmeasured economic and political factors [1].

In this work, we considered the problem of causal discovery in setups with such challenges, particularly under two settings: (1) in the presence of measurement error, and (2) with unobserved parentless causes. Unlike previous work, our proposed SEM-ME allows for applications containing a mixture of variables measured exactly (without error) or measured with error. We demonstrated a mapping between these two models that preserves their mixing matrix. Based on this mapping, we derived identifiability results for both models under different faithfulness assumptions, and proposed

---

[4]Our code is available at: `https://github.com/Yuqin-Yang/SEM-ME-UR`.

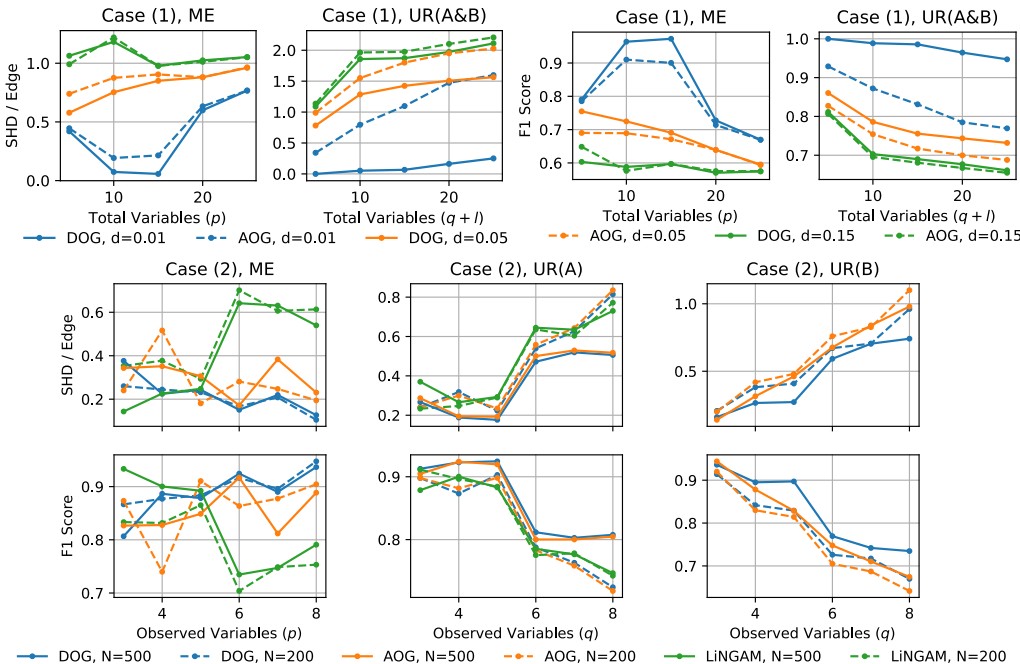

Figure 4: Simulation results for Case (1), noisy mixing matrix (with noise variance $\mathtt{d}^2$), and Case (2), raw observational data (with sample size $\mathtt{N}\cdot p$ for ME and $\mathtt{N}\cdot q$ for UR). In Case (1), x-axis corresponds to $|\mathcal{Y} \cup \mathcal{Z}|$ in SEM-ME, and $|\mathcal{H} \cup \mathcal{X}|$ in SEM-UR. In Case (2), x-axis corresponds to $|\mathcal{Y} \cup \mathcal{U}|$ in SEM-ME, and $|\mathcal{X}|$ in SEM-UR. We compare the recovery of both adjacency matrices **A** and **B** (cf. Equation (4)) for SEM-UR. Lower SHD/Edge value and higher F1 score indicate better performances.

structure learning algorithms. Our results further implied conditions for unique identifiability of the structure in SEM-UR. These conditions do not pose any restrictions on the graphical model, and hence significantly relax the existing graphical conditions in the literature [19, 1, 25, 4].

Our results have several implications in the literature of causal discovery including the following: The mapping proposed in our work between linear SEM-ME and SEM-UR allows us to translate identifiability results for one model to the other. We note that linear SEM-UR has been widely studied in the literature, while only a few works considered linear SEM-ME. Therefore, an important implication of our work is that the introduced mapping can be utilized to fill the gaps for the less studied model. Another important implication of our result is that it can be used for evaluating new algorithms: We showed that under two different faithfulness assumptions, the model can only be identified up to AOG and DOG equivalence classes. This result provides the extent of identifiability, and can serve as a basis for characterizing theoretical guarantees such as consistency of new algorithms. Finally, we showed in Proposition 4 that under our faithfulness assumptions, the true generating model is always sparser than any other model in the same AOG equivalence class (of the ground-truth) but does not belong to the DOG equivalence class. This serves as a ground for positing sparsity assumptions that appear frequently in the literature without rigorous justification.

Similar to other causal discovery methods which are based on the mixing matrix, the performance of our proposed recovery algorithms relies on the accuracy of the utilized mixing matrix estimation approach. Hence, the recovered structure should be interpreted with caution if the mixing matrix estimation approach is unreliable. Devising more accurate approaches for estimating the mixing matrix, as well as extending the method proposed in this work to non-linear models are important directions of future research.

## Acknowledgments and Disclosure of Funding

Negar Kiyavash's research was in part supported by the Swiss National Science Foundation under NCCR Automation, grant agreement 51NF40_180545 and Swiss SNF project 200021_204355 /1. Kun Zhang was partially supported by NIH under Contract R01HL159805, by the NSF-Convergence Accelerator Track-D award #2134901, by a grant from Apple Inc., and by a grant from KDDI Research Inc. Ilya Shpitser was supported by grants ONR N00014-21-1-2820, NSF 2040804, NSF CAREER 1942239 and NIH R01 AI127271-01A1.

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
