# Supplementary Material

## A   Further discussion on ancestral ordered grouping

First, we compare our ancestral ordered grouping (AOG) in Definition 4 with the ordered grouping proposed in [29]. Next, we compare our AOG to the equivalence classes for identifiability of a linear SEM with latent variables under conventional faithfulness, proposed in [19]. Finally, we present the details of our AOG recovery algorithm.

### A.1   Comparison of AOG to previous work

In the following, we present an equivalent definition of the ordered grouping proposed in [29], which is closer in spirit to our AOG definition. As we mentioned earlier, the model proposed in [29] assumes that all the underlying variables are measured with error (unobserved), i.e., $\mathcal{Y}$ in Equation (1) is an empty set. Subsequently, a `u-leaf` node is an unobserved underlying variable with no children in $\mathcal{Z}$ and an `nu-leaf` node is an unobserved underlying variable with at least one child in $\mathcal{Z}$.

**Definition 8 (Ordered group decomposition (OGD) [29])**  *The ordered group decomposition of a linear SEM-ME is a partition of $\mathcal{Z}$ into distinct sets. This partition is described as follows:*

*(1)  Select a causal order $k$ among `nu-leaf` nodes in $\mathcal{Z}$ that is consistent with the generating model. Assign each `nu-leaf` node to a distinct ordered group.*

*(2)  For each `u-leaf` node $Z_j \in \mathcal{Z}$, assign $Z_j$ to the same group as its parent with the largest index in $k$.*

The ordered grouping by [29] depends on the selected causal order and hence might not be unique, since there can be more than one causal order consistent with the generating model. Subsequently, this ordered grouping does not permit characterizing the extent of identifiability under conventional faithfulness alone, i.e., the equivalence classes of the generating model in Theorem 2. In contrast, our AOG is a more refined grouping than Definition 8 and returns a unique partition that does not depend on the selection of the causal order. Further, our AOG characterizes the extent of identifiability under conventional faithfulness (cf. Theorem 2). The following example illustrates the difference between the ordered grouping in [29] and our AOG.

**Example 3**  *Consider a collider structure over three underlying variables in a linear SEM-ME: $Z_1 \rightarrow Z_3 \leftarrow Z_2$, where $Z_3$ is a `u-leaf` node. There are two possible causal orders that correspond to this structure, namely (1,2,3) and (1,3,2). Hence, based on the ordered grouping definition of [29], there are two possible corresponding partitions, namely, $\{\{Z_1\}, \{Z_2, Z_3\}\}$ and $\{\{Z_2\}, \{Z_1, Z_3\}\}$. However, following our definition of AOG, i.e., Definition 4, the AOG is $\{\{Z_1\}, \{Z_2\}, \{Z_3\}\}$, which is a refined partition of either of the two partitions by [29]. Further, the AOG of the model can be recovered based on (the support of) the mixing matrix $\mathbf{W}^{ME}$: $Z_3$ includes two noise terms, while $Z_1$ and $Z_2$ each includes one.*

Next, we show the connection between our AOG and the observed descendant sets in [19] proposed for a linear SEM with latent variables. The "observed descendant set" of a (latent or observed) variable $V_i$ is defined as the set of all observed descendants of $V_i$, including $V_i$ itself if it is observed. The following proposition states the equivalence of the two notions in a linear SEM-UR.

**Proposition 5**  *In a linear SEM-UR, two variables are in the same ancestral ordered group if and only if they have the same observed descendant set.*

The authors of [19] showed that using "observed descendant sets", the causal order among the observed variables can be identified under conventional faithfulness assumption (i.e., Assumption 3*(a)*). Further, the total causal effects among observed variables can be estimated. However, the model in [19] allows for latent variables which have parents (i.e., not roots) and in general, the full structure cannot be identified. In contrast, our AOG equivalence class characterization (Theorem 2) extends the result in [19]. Specifically, we can recover the causal order among the ancestral ordered groups, where each ancestral ordered group has at most one observed variable. Moreover, we can estimate the exact structure and the coefficients of every model in the AOG equivalence class.

---

**Algorithm 1:** Recovering ancestral ordered grouping in linear SEM-ME.

---

**Input:** Recovered mixing matrix $\mathbf{W}^{ME}$.

1  Calculate the number of non-zero entries in each row of $\mathbf{W}^{ME}$. Denote the vector of these numbers as $\mathbf{n}$, where each entry in $\mathbf{n}$ corresponds to a row in $\mathbf{W}^{ME}$.

2  Initialize $\tilde{\mathbf{W}} = \mathbf{W}^{ME}$.

3  **while** $\tilde{\mathbf{W}}$ *is not empty* **do**

4      Find a row in $\tilde{\mathbf{W}}$ that contains only one non-zero entry, and has the smallest corresponding value in $\mathbf{n}$. If there are multiple such rows, randomly select one. Denote the selected row as $\mathbf{w}$, and its corresponding value in $\mathbf{n}$ as $n_0$.

5      Consider the rows in $\tilde{\mathbf{W}}$ with the same support as $\mathbf{w}$ (including $\mathbf{w}$ itself). Denote the set of variables that correspond to these rows as $\mathcal{Z}_W$.

6      Assign all variables in $\mathcal{Z}_W$, with the same corresponding value as $n_0$ in $\mathbf{n}$, to a single ordered group.

7      Assign each of the remaining variables in $\mathcal{Z}_W$ to a separate ordered group.

8      Remove from $\tilde{\mathbf{W}}$ the rows corresponding to the variables in $\mathcal{Z}_W$, and the column containing the corresponding non-zero entries in these rows.

**Output:** AOG of the SEM-ME.

---

### A.2  AOG recovery algorithm

Based on Assumption 1, we can recover $\mathbf{W}^{ME}$ (resp. $\mathbf{W}^{UR}$) from the observed data. The following property can be implied from the definition of AOG, which shows that under conventional faithfulness assumption, we can identify the AOG of the model only based on the support of the mixing matrix.

**Proposition 6** *(a) Under Assumption 2*(a)*, two variables in a SEM-ME belong to the same ancestral ordered group if and only if the two rows in $\mathbf{W}^{ME}$ corresponding to these variables have the same support. (b) Under Assumption 3*(a)*, two variables in a SEM-UR belong to the same ancestral ordered group if and only if the two columns in $\mathbf{W}^{UR}$ corresponding to the exogenous noise terms of these variables have the same support.*

Equipped with this proposition, Algorithm 1 shows how to recover the AOG in Definition 4 from $\mathbf{W}^{ME}$. The Algorithm first randomly chooses a row in $\mathbf{W}^{ME}$ with one non-zero entry and finds all other rows with the same support, and puts all the corresponding variables in a single ancestral ordered group. Next, it removes the selected rows and the column that contains the corresponding non-zero entry. In the remaining matrix, say $\tilde{\mathbf{W}}$, the Algorithm again chooses a row $\mathbf{w}$ with one non-zero entry, but has the smallest number of non-zero entries in $\mathbf{W}^{ME}$. Then, it selects all other rows in $\tilde{\mathbf{W}}$ with the same support and the same number of non-zero entries in $\mathbf{W}^{ME}$, as $\mathbf{w}$. The rows in $\tilde{\mathbf{W}}$ with exactly one non-zero entry and having the same (fewest) number of non-zero entries in $\mathbf{W}^{ME}$ correspond to an `nu-leaf` node and its direct children (with the same support in $\mathbf{W}^{ME}$), and hence they go together into the same group. The remaining rows with more non-zero entries in $\mathbf{W}^{ME}$ are `u-leaf` nodes that go to separate distinct groups. Finally, this procedure is repeated until all variables are assigned to ancestral ordered groups.

Since Algorithm 1 is based on the mixing matrix, it follows from the constructed mapping in Theorem 1 that the same Algorithm can be applied to recover the AOG of a linear SEM-UR, by replacing "rows" with "columns" and vice versa, in the Algorithm.

## B  SEM-ME and SEM-UR faithfulness assumptions

We first present the formal statement of SEM-ME and SEM-UR faithfulness assumptions in Appendix B.1. We also provide examples of violations of faithfulness. In Appendix B.2, we present an equivalent representation of SEM-ME and SEM-UR faithfulness, using the mixing matrix. Next, in Appendix B.3, we show that violation of faithfulness assumptions is a measure-zero event. Finally, we present the relation between bottleneck faithfulness [1] and SEM-UR faithfulness assumptions in Appendix B.4.

### B.1  Formal statement of faithfulness assumption

For a variable $V_i$, we define the ancestor set, $An(V_i)$ (resp. the descendant set, $De(V_i)$) as the sets of variables that have directed paths to $V_i$ (resp. from $V_i$); both excluding $V_i$ itself. Note that in the literature, some works use the convention $V_i \in An(V_i)$ and $V_i \in De(V_i)$. We do not use this convention here.

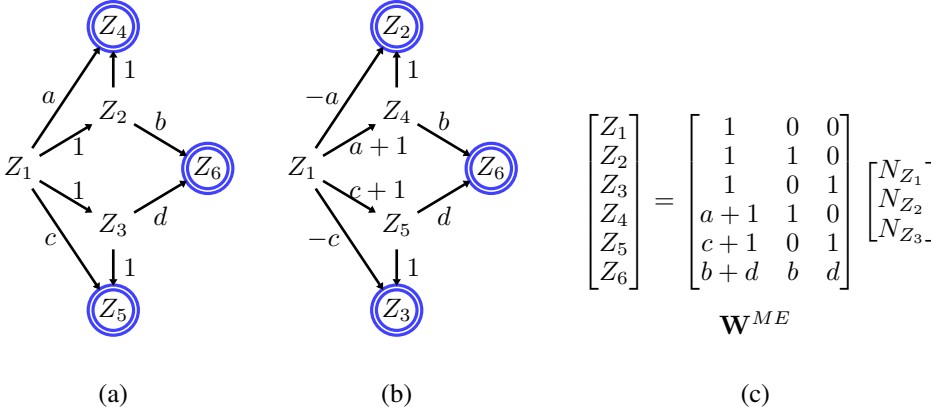

Figure 5: Example explaining Assumption 2*(b)*. (a) The ground-truth model of a linear SEM-ME. (b) An alternative linear SEM-ME. (c) If $ab + cd = 0$, then both models have the same mixing matrix $\mathbf{W}^{ME}$.

For SEM-ME, define the possible parent set of a variable $V_i \in \mathcal{Z} \cup \mathcal{Y}$, as the union of the ancestor set of $V_i$, $An(V_i)$, and the set of u-leaf nodes whose parents are subsets of $An(V_i)$. For any set $\mathcal{V} \subseteq \mathcal{Z} \cup \mathcal{Y}$, define $\text{TE}(\mathcal{V}, V_i)$ as the vector whose entries represent the total causal effects (i.e., sum of products of path coefficients) of the elements of $\mathcal{V}$ on $V_i$. We define the total causal effect of a variable on itself to be 1. Similarly, for SEM-UR, define the possible children set of $V_i \in \mathcal{H} \cup \mathcal{X}$ as the union of the descendant set of $V_i$, $De(V_i)$, and the set of latent variables whose children sets are subsets of $De(V_i)$. For any set $\mathcal{V} \subseteq \mathcal{H} \cup \mathcal{X}$, define $\text{TC}(V_i, \mathcal{V})$ as the vector whose entries represent the total causal effects of $V_i$ on the elements of $\mathcal{V}$.

**Assumption 2 (SEM-ME faithfulness)** *(a) The total causal effect of an underlying variable $V_i \in \mathcal{Z} \cup \mathcal{Y}$ on its descendant $V_j \in \mathcal{Z} \cup \mathcal{Y}$ is not equal to zero.*
*(b) For each variable $V_i \in \mathcal{Z} \cup \mathcal{Y}$, (b1) $\text{TE}(An(V_i), V_i)$ is linearly independent of any $k \le |Pa(V_i)|$ vectors in $\{\text{TE}(An(V_i), V) : V \text{ is a possible parent of } V_i\}$, except for the case when $k = |Pa(V_i)|$ and for each u-leaf node $V_l$ that corresponds to one of these $k$ vectors, $Pa(V_l) \subseteq Pa(V_i)$; (b2) If $V_i$ is a u-leaf node, then for each parent $V_j$ of $V_i$, $\text{TE}(An(V_i) \setminus \{V_j\}, V_i)$ is linearly independent of any $k \le |Pa(V_j) \cup Pa(V_i)| - 1$ vectors in $\{\text{TE}(An(V_i) \setminus \{V_j\}, V) : V \text{ is a possible parent of } V_j\}$, except for the case when $k = |Pa(V_j) \cup Pa(V_i)| - 1$ and for each u-leaf node $V_l$ that corresponds to one of these $k$ vectors, $Pa(V_l) \subseteq Pa(V_j) \cup Pa(V_i)$.*

**Assumption 3 (SEM-UR faithfulness)** *(a) The total causal effect of an observed (latent) variable $X_i \in \mathcal{X}$ $(H_i \in \mathcal{H})$ on its descendant $X_j \in \mathcal{X}$ is not equal to zero.*
*(b) For each variable $V_i \in \mathcal{X} \cup \mathcal{H}$, (b1) $\text{TC}(V_i, De(V_i))$ is linearly independent of any $k \le |Ch(V_i)|$ vectors in $\{\text{TC}(V, De(V_i)) : V \text{ is a possible child of } V_i\}$, except for the case when $k = |Ch(V_i)|$ and for each latent variable $H_l$ that corresponds to one of these $k$ vectors, $Ch(H_l) \subseteq Ch(V_i)$; (b2) If $V_i$ is a latent variable, then for each child $X_j$ of $V_i$, $\text{TC}(V_i, De(V_i) \setminus \{X_j\})$ is linearly independent of any $k \le |Ch(X_j) \cup Ch(V_i)| - 1$ vectors in $\{\text{TC}(V, De(V_i) \setminus \{X_j\}) : V \text{ is a possible child of } X_j\}$, except for the case when $k = |Ch(X_j) \cup Ch(V_i)| - 1$ and for each latent variable $H_l$ that corresponds to one of these $k$ vectors, $Ch(H_l) \subseteq Ch(X_j) \cup Ch(V_i)$.*

We explain the aforementioned faithfulness assumptions using the following examples, which investigate various cases of violation of SEM-ME faithfulness, and show that our identifiability result in Theorem 3 may no longer hold under such violation. In Appendix B.3, we show that violation of SEM-ME faithfulness is a measure-zero event.

**Example 4** *Consider the linear SEM-ME generated by the graph in Figure 2(a).[5] The mixing matrix is given in Figure 2. We have $\text{TE}(\{Z_1, Z_2\}, Z_4) = [a(b + c); b + c] = (b + c)\text{TE}(\{Z_1, Z_2\}, Z_2)$. This implies that $\text{TE}(An(Z_4) \setminus \{Z_3\}, Z_4)$ is linearly dependent on one vector $\text{TE}(An(Z_4) \setminus \{Z_3\}, Z_2)$ where $Z_2$ is a possible parent of $Z_3$. Hence the model violates Assumption 2(b2).*

*According to Definition 6, the DOG of the ground truth model (Figure 2(a)) is $\{\{Z_1\}, \{Z_2\}, \{Z_3, Z_4\}\}$. Because either $Z_3$ or $Z_4$ can be the center of the last group, the DOG equivalence class should include both the ground-truth and the model represented by Figure 2(b). However, due to violation of Assumption 2(b2), the model represented by Figure 2(b), does not belong to the same DOG equivalence class as the ground-truth, and hence can be distinguished from the ground truth. Therefore, our identifiability result in Theorem 3 does not hold because of violation of faithfulness.*

---

[5]This example was stated in Section 4.1, and we repeat it here using the notation $\text{TE}(\cdot, \cdot)$ in the formal statement of Assumption 2*(b)*.

**Example 5** *Consider the linear SEM-ME generated by the graph in Figure 5(a). The mixing matrix is given in Figure 5(c). We have* $\mathtt{TE}(An(Z_6), Z_6) = [b+d; b; d]$, $\mathtt{TE}(An(Z_6), Z_4) = [a+1; 1; 0]$, *and* $\mathtt{TE}(An(Z_6), Z_5) = [c + 1; 0; 1]$. *If* $ab + cd = 0$, *then* $\mathtt{TE}(An(Z_6), Z_6)$ *can be written as* $b\mathtt{TE}(An(Z_6), Z_4) + d\mathtt{TE}(An(Z_6), Z_5)$. *That is,* $\mathtt{TE}(An(Z_6), Z_6)$ *can be written as the linear combination of two vectors; however, the variables corresponding to these two vectors, i.e.,* $Z_4, Z_5$, *are both* `u-leaf` *nodes and have a parent* $Z_1$ *that is not a parent of* $Z_6$. *Therefore the model violates Assumption 2(b1). Note that* $ab + cd = 0$ *is a measure-zero event if all parameters are randomly and independently drawn from continuous distributions.*

*Note that each of the six variables in the ground-truth model belongs to a separate direct ordered group. This means that the DOG equivalent class only includes the ground-truth. However, if* $ab + cd = 0$, *then the linear SEM-ME generated by the graph in Figure 5(b) is indistinguishable from the ground truth, since it has the same mixing matrix and the same structure (i.e., the two graphs are isomorphic). Once again, our identifiability result in Theorem 3 does not hold because of violation of faithfulness.*

## B.2 Equivalent matrix form for faithfulness assumptions

In this section we present an equivalent matrix representation of Assumptions 2*(b)* and 3*(b)*. For each variable $V_i$ in SEM-ME, define $Supp(V_i)$ as the support of the row in $\mathbf{W}^{ME}$ corresponding to $V_i$: $Supp(V_i) = \{N_{V_j}|j : [\mathbf{W}^{ME}]_{ij} \neq 0\}$. For each variable $V_i$ in SEM-UR, define $Supp(N_{V_i})$ as the support of the column in $\mathbf{W}^{UR}$ corresponding to $N_{V_i}$: $Supp(N_{V_i}) = \{V_j|j : [\mathbf{W}^{UR}]_{ji} \neq 0\}$.

Under this formulation, a variable $V_j$ in a SEM-ME is a possible parent of $V_i$ if and only if $V_j \neq V_i$, and $Supp(V_j)$ is a subset of $Supp(V_i) \setminus \{N_{V_i}\}$, i.e., $Supp(V_i)$ excluding the exogenous noise term $N_{V_i}$ if $V_i$ is an `nu-leaf` node [28]. A variable $V_j$ in a SEM-UR is a possible child of $V_i$ if and only if $V_j \neq V_i$ and $Supp(N_{V_j})$ is a subset of $Supp(N_{V_i}) \setminus \{V_i\}$, i.e., $Supp(V_i)$ excluding $V_i$ itself if $V_i$ is observed.

**Proposition 7** *Assumptions 2(b) and 3(b) have the following equivalent matrix form:*

**Assumption 2(b):** *For a variable* $V_i \in \mathcal{Z} \cup \mathcal{Y}$, *(b1) denote* $\mathbf{W}_i^{ME}$ *as the submatrix of* $\mathbf{W}^{ME}$ *with rows corresponding to the possible parents of* $V_i$, *and columns corresponding to* $Supp(V_i) \setminus \{N_{V_i}\}$. *Then the vector* $\mathbf{W}^{ME}[V_i, Supp(V_i) \setminus \{N_{V_i}\}]$ *cannot be written as any linear combination of* $k \leq |Pa(V_i)|$ *rows in* $\mathbf{W}_i^{ME}$, *except for the case when* $k = |Pa(V_i)|$ *and for each* `u-leaf` *node* $V_l$ *that corresponds to one of these* $k$ *rows,* $Pa(V_l) \subseteq Pa(V_i)$.
*(b2) If* $V_i$ *is a* `u-leaf` *node, then for each parent* $V_j$ *of* $V_i$, *denote* $\mathbf{W}_{ij}^{ME}$ *as the submatrix of* $\mathbf{W}^{ME}$ *with rows corresponding to the possible parents of* $V_j$, *and columns corresponding to* $Supp(V_i) \setminus \{N_{V_j}\}$. *Then* $\mathbf{W}^{ME}[V_i, Supp(V_i) \setminus \{N_{V_j}\}]$ *cannot be written as any linear combination of* $k \leq |Pa(V_i) \cup Pa(V_j)| - 1$ *rows in* $\mathbf{W}_{ij}^{ME}$, *except for the case when* $k = |Pa(V_i) \cup Pa(V_j)| - 1$ *and for each* `u-leaf` *node* $V_l$ *that corresponds to one of these* $k$ *rows,* $Pa(V_l) \subseteq Pa(V_i) \cup Pa(V_j)$.

**Assumption 3(b):** *For a variable* $V_i \in \mathcal{H} \cup \mathcal{X}$, *(b1) denote* $\mathbf{W}_i^{UR}$ *as the submatrix of* $\mathbf{W}^{UR}$ *with columns corresponding to the exogenous noise terms of the possible children of* $V_i$, *and rows corresponding to* $Supp(N_{V_i}) \setminus \{V_i\}$. *Then the vector* $\mathbf{W}^{UR}[Supp(N_{V_i}) \setminus \{V_i\}, N_{V_i}]$ *cannot be written as any linear combination of* $k \leq |Ch(V_i)|$ *columns in* $\mathbf{W}_i^{UR}$, *except for the case when* $k = |Ch(V_i)|$ *and for each latent variable* $H_l$ *whose exogenous noise term corresponds to one of these* $k$ *columns,* $Ch(H_l) \subseteq Ch(V_i)$.
*(b2) If* $V_i$ *is a latent variable, then for each child* $X_j$ *of* $V_i$, *denote* $\mathbf{W}_{ji}^{UR}$ *as the submatrix of* $\mathbf{W}^{UR}$ *with columns corresponding to the exogenous noise terms of the possible children of* $X_j$, *and rows corresponding to* $Supp(N_{V_i}) \setminus \{X_j\}$. *Then* $\mathbf{W}^{UR}[Supp(N_{V_i}) \setminus \{X_j\}, N_{V_i}]$ *cannot be written as any linear combination of* $k \leq |Ch(V_i)| - 1$ *columns in* $\mathbf{W}_{ji}^{UR}$, *except for the case when* $k = |Ch(X_j) \cup Ch(V_i)| - 1$ *and for each latent variable* $H_l$ *whose exogenous noise term corresponds to one of these* $k$ *columns,* $Ch(H_l) \subseteq Ch(X_j) \cup Ch(V_i)$.

## B.3 Violation of our faithfulness assumptions is a measure-zero event

In the following, we show that Assumption 2 is violated with probability zero if all model coefficients are randomly and independently drawn from continuous distributions. Same argument holds for Assumption 3, and hence the proof is omitted. Note that Assumption 2*(a)* is identical to conventional faithfulness in linear causal models, the violation of which is known to be a measure-zero event [15, 24]. Thus, it suffices to show that violation of Assumption 2*(b)* is a measure-zero event. We do so for both parts of Assumption 2*(b)*.

Assumption 2*(b1)*: Let $\mathcal{T}_i$ denote the set of vectors $\{\mathtt{TE}(An(V_i), V) : V \text{ is a possible parent of } V_i\}$. Recall that the total effects of $An(V_i)$ on $V_i$, i.e., $\mathtt{TE}(An(V_i), V_i)$, can be written as a linear combination of the total effects of $An(V_i)$ on parents of $V_i$, i.e., $\mathcal{P}_i = \{\mathtt{TE}(An(V_i), V) : V \in Pa(V_i)\}$:

$$\mathtt{TE}(An(V_i), V_i) = \sum_{V_j \in Pa(V_i)} a_{ij}\mathtt{TE}(An(V_i), V_j). \tag{8}$$

Further, $\mathcal{A}_i = \{\text{TE}(An(V_i), V) : V \in An(V_i)\}$ constitutes a basis for $\text{span}(\mathcal{T}_i)$: Vectors in $\mathcal{A}_i$ are linearly independent (because each ancestor has its exogenous noise term) and every vector in $\mathcal{T}_i$ (which corresponds to a possible parent of $V_i$) can be written as a linear combination of vectors in $\mathcal{A}_i$. Note that $\mathcal{P}_i \subseteq \mathcal{A}_i$. Therefore, if Assumption 2*(b1)* is violated, then $\text{TE}(An(V_i), V_i)$ can either be (i) linearly dependent on $k < |Pa(V_i)|$ vectors in $\mathcal{T}_i$, or (ii) linearly dependent on $|Pa(V_i)|$ vectors in $\mathcal{T}_i$ and at least one of these vectors does not belong to $\text{span}(\mathcal{P}_i)$. Both of these events are of measure zero, when all $\{a_{ij}|j : V_j \in Pa(V_i)\}$ in Equation (8) are randomly and independently selected from continuous distributions. This follows because if we represent $\text{TE}(An(V_i), V_i)$ and all vectors in $\mathcal{T}_i$ as linear combinations of the vectors in $\mathcal{A}_i$, then case (i) corresponds to a solution of a linear system with $k$ variables and at least $|Pa(V_i)|$ equations (constraints), and case (ii) corresponds to a solution of a linear system with $|Pa(V_i)|$ variables and at least $|Pa(V_i)| + 1$ equations.

Assumption 2*(b2)*: Similarly, we first show that for any parent $V_j$ of $V_i$, $\text{TE}(An(V_i) \setminus \{V_j\}, V_i)$ can be written as the linear combination of $|Pa(V_i) \cup Pa(V_j)| - 1$ linearly independent vectors in $\mathcal{T}_i$ with probability one. To show this, we first note that Equation (8) holds for any subvector of $\text{TE}(An(V_i), V_i)$. Therefore,

$$\text{TE}(An(V_i) \setminus \{V_j\}, V_i) = \sum_{V_{j'} \in Pa(V_i)} a_{ij'} \text{TE}(An(V_i) \setminus \{V_j\}, V_{j'}).$$

Further, similar to Equation (8), $\text{TE}(An(V_j), V_j)$ can also be written as a linear combination of the vectors in $\{\text{TE}(An(V_j), V_k) : V_k \in Pa(V_j)\}$. Since the total causal effect from any variable in $An(V_i) \setminus (An(V_j) \cup \{V_j\})$ to $V_j$ or any parent of $V_j$ is zero, we have

$$\text{TE}(An(V_i) \setminus \{V_j\}, V_j) = \sum_{V_k \in Pa(V_j)} a_{jk} \text{TE}(An(V_i) \setminus \{V_j\}, V_k).$$

Combining these two equations, we have

$$
\begin{aligned}
\text{TE}(An(V_i) \setminus \{V_j\}, V_i) =& \sum_{V_{j'} \in Pa(V_i) \setminus \{V_j\}} a_{ij'} \text{TE}(An(V_i) \setminus \{V_j\}, V_{j'}) \\
&+ \sum_{V_k \in Pa(V_j)} a_{ij} a_{jk} \text{TE}(An(V_i) \setminus \{V_j\}, V_k).
\end{aligned}
\tag{9}
$$

Note that the right hand side of Equation (9) includes $|Pa(V_i) \cup Pa(V_j)| - 1$ vectors, and for each vector that is included in both summations, the coefficients cancel out each other with probability zero.

Let $\mathcal{T}_{ji}$, $\mathcal{A}_{ji}$, $\mathcal{P}_{ji}$ denote the set of vectors $\{\text{TE}(An(V_i) \setminus \{V_j\}, V) : V$ is a possible parent of $V_j\}$, $\{\text{TE}(An(V_i) \setminus \{V_j\}, V) : V \in An(V_j)\}$, $\{\text{TE}(An(V_i) \setminus \{V_j\}, V) : V \in Pa(V_j) \cup Pa(V_i) \setminus \{V_j\}\}$, respectively. Similarly, $\mathcal{A}_{ji}$ constitutes a basis of $\text{span}\mathcal{T}_{ji}$. If Assumption 2*(b2)* is violated, then all parents of $V_i$ are ancestors of $V_j$, otherwise $\text{TE}(An(V_i) \setminus \{V_j\}, V_i)$ cannot belong to $\text{span}(\mathcal{T}_{ji})$. This means that all vectors on the right hand side of Equation (9), i.e., $\mathcal{P}_{ji}$, belong to $\mathcal{A}_{ji}$. Further, $\text{TE}(An(V_i) \setminus \{V_j\}, V_i)$ can either be (i) linearly dependent on $k < |Pa(V_i) \cup Pa(V_j)| - 1 = |\mathcal{P}_{ji}|$ vectors in $\mathcal{T}_{ji}$, or (ii) linearly dependent on $|Pa(V_i) \cup Pa(V_j)| - 1$ vectors in $\mathcal{T}_{ji}$ and at least one of these vectors does not belong to $\text{span}(\mathcal{P}_{ji})$. Both of these events are measure zero, when all $\{a_{ij'}|j' : V_{j'} \in Pa(V_i), j' \neq V_j\} \cup \{a_{jk}|k : V_k \in Pa(V_j)\}$ in Equation (9) are randomly and independently selected from continuous distributions.

## B.4 Bottleneck faithfulness implies SEM-UR faithfulness

We show that our SEM-UR faithfulness is weaker than Bottleneck faithfulness in [1], in the sense that more models satisfy SEM-UR faithfulness. Bottleneck faithfulness is defined as follows:

**Condition 3 (Bottleneck faithfulness)** *Define a bottleneck $B$ from $\mathcal{J}$ to $\mathcal{K}$ as a set of variables where any directed path from $j \in \mathcal{J}$ to $k \in \mathcal{K}$ includes some variables in $B$, and a minimal bottleneck $B$ from $\mathcal{J}$ to $\mathcal{K}$ as the bottleneck with the smallest size. Then for every $\mathcal{J} \subseteq \mathcal{H} \cup \mathcal{X}$, $\mathcal{K} \subseteq \mathcal{X}$, if $B$ is a minimal bottleneck from $\mathcal{J}$ to $\mathcal{K}$, then $\text{Rank}(\mathbf{W}_{\mathcal{K}}^{\mathcal{J}}) = |B|$, where $\mathbf{W}_{\mathcal{K}}^{\mathcal{J}}$ is the submatrix of $\mathbf{W}^{U\bar{R}}$ representing the total causal effect from variables in $J$ to variables in $K$.*

"Bottleneck faithfulness implies SEM-UR faithfulness": It has been shown in [1] that Condition 3 implies Assumption 3*(a)*. For Assumption 3*(b)*, it suffices to show that if either Assumption 3*(b1)* or *(b2)* is violated, then Condition 3 is violated.

*Assumption 3(b1)*: Suppose $V_i$ violates Assumption *(b1)*. Then $\text{TC}(V_i, De(V_i))$ is either: (i) linearly dependent on $k < |Ch(V_i)|$ vectors in the set $\mathcal{T}_i = \{\text{TC}(V, De(V_i)) : V$ is a possible child of $V_i\}$, or (ii) linearly dependent on $k = |Ch(V_i)|$ vectors in $\mathcal{T}_i$ and the children set of at least one latent variable $H_j$ corresponding to these vectors is not a subset of $|Ch(V_i)|$. Without loss of generality, suppose $k$ is minimal, i.e, $\text{TC}(V_i, De(V_i))$ is linearly independent of any $l$ vectors in $\mathcal{T}_i$ for any $l < k$. Denote the set of possible children of $V_i$ corresponding to these vectors as $\mathcal{P}$. Consider the set $\mathcal{J} = \mathcal{P} \cup \{V_i\}$, and $\mathcal{K} = Des(V_i)$.[6] Then $\text{Rank}(\mathbf{W}_{\mathcal{K}}^{\mathcal{J}}) = |\mathcal{P}| = k$,

---

[6]Recall that we do not include $V_i$ itself in $Des(V_i)$. This is different from [1] where they include $V_i$.

because (i) $\mathbf{W}_{\mathcal{K}}^{V_i}$ can be written as a linear combination of the columns in $\mathbf{W}_{\mathcal{K}}^{\mathcal{P}}$, and (ii) columns in $\mathbf{W}_{\mathcal{K}}^{\mathcal{P}}$ are linearly independent (otherwise $k$ is not minimal).

In the following we show that any bottleneck from $\mathcal{J}$ to $\mathcal{K}$ is at least of size $k + 1$. First, any bottleneck from $\mathcal{P}$ to $\mathcal{K}$ is at least of size $|\mathcal{P}| = k$. This is because if there exists a bottleneck of size $k - 1$, then according to Proposition 1 of [1], $\mathrm{Rank}(\mathbf{W}_{\mathcal{K}}^{\mathcal{J}}) \leq k - 1$. This contradicts the minimality of $k$. Next, for any bottleneck $B$ from $\mathcal{P}$ to $\mathcal{K}$ with size $k$: (i) If $k < |Ch(V_i)|$, then $B$ will cover at most $k$ children of $V_i$. (ii) If $k = |Ch(V_i)|$. Note that there exists at least one child of $H_j$, denoted by $X_k$, that is not a child of $V_i$. Therefore, to block the edge $H_j \to X_k$, $B$ must include at least one variable between $H_j$ and $X_k$. This implies that $B$ includes at most $|Ch(V_i)| - 1$ children of $V_i$.

In either case, there is at least one directed edge from $V_i$ to one of its children that cannot be blocked by $B$. Therefore, the minimal bottleneck from $\mathcal{J}$ to $\mathcal{K}$ is at least of size $k + 1$, which violates bottleneck faithfulness condition.

*Assumption 3(b2)*: Similarly, suppose $V_i$ and its child $X_j$ violates Assumption *(b2)*. Then $\mathrm{TC}(V_i, De(V_i) \setminus \{X_j\})$ is either: (i) linearly dependent on $k < |Ch(V_i) \cup Ch(X_j)| - 1$ vectors in the set $\mathcal{T}_{ji} = \{\mathrm{TC}(V, De(V_i) \setminus \{X_j\}) : V$ is a possible child of $X_j\}$, or (ii) linearly dependent on $k = |Ch(V_i) \cup Ch(X_j)| - 1$ vectors in $\mathcal{T}_{ji}$, and the children set of at least one latent variable $H_k$ corresponding to these vectors is not a subset of $Ch(V_i) \cup Ch(X_j)$.

Suppose $k$ is minimal, and denote the set of possible children corresponding to these $k$ vectors as $\mathcal{P}$. Consider the set $\mathcal{J} = \mathcal{P} \cup \{V_i\}$ and $\mathcal{K} = Des(V_i) \setminus \{X_j\}$. Then $\mathrm{Rank}(\mathbf{W}_{\mathcal{K}}^{\mathcal{J}}) = |\mathcal{P}| = k$. However, any bottleneck from $\mathcal{J}$ to $\mathcal{K}$ is at least of size $k + 1$. This is because: (i) Any bottleneck from $\mathcal{P}$ to $\mathcal{K}$ is at least of size $|\mathcal{P}| = k$; (ii) For any bottleneck $B$ from $\mathcal{P}$ to $\mathcal{K}$ with size $k$, there are $|Ch(V_i) \cup Ch(X_j) \setminus \{X_j\}|$ variables in $\mathcal{K}$ that is a child of either $V_i$ or $X_j$. Therefore there is at least one directed edge from $V_i$ or $X_j$ to one of their children (excluding $X_j$) that cannot be blocked by $B$. (If $|B| = |Ch(V_i) \cup Ch(X_j)| - 1$, then there is at least one variable in $B$ that blocks the edge from $H_k$ to its child that does not belong to $Ch(V_i) \cup Ch(X_j)$.) Therefore bottleneck faithfulness condition is violated.

"SEM-UR faithfulness does not imply bottleneck faithfulness": In the following we will provide an example which satisfies Assumption 3 but violates Condition 3. This example is introduced in [1]. Consider the following linear SEM-UR involving three latent variables $H = [H_1, H_2, H_3]$ and four observed variables $X = [X_1, X_2, X_3, X_4]$:

$$H = N_H, \quad X = \begin{bmatrix} 0 & 1 & -1 \\ 2 & 2 & 0 \\ 3 & 3 & 0 \\ 4 & 0 & 4 \end{bmatrix} H + \mathbf{0}X + N_X.$$

It has been shown in [1] that the minimal bottleneck from $\mathcal{H} = \{H_1, H_2, H_3\}$ to $\mathcal{X} = \{X_1, X_2, X_3, X_4\}$ is $\mathcal{H}$ with $|\mathcal{H}| = 3$, while $\mathrm{Rank}(\mathbf{W}_{\mathcal{X}}^{\mathcal{H}}) = 2$. Therefore the model violates Condition 3. On the contrary, the model satisfies Assumption 3. This is because the possible children set of each observed variable is empty set (meaning that *(b2)* is satisfied), and the possible children set of each latent variable only includes its children (meaning that *(b1)* is satisfied).

## C  Proofs of the main results

### C.1  Proof of Theorem 1

We first observe that in a SEM-ME, whether a non-leaf underlying variable $V_i$ is observed (i.e., $V_i \in \mathcal{Y}$) or measured with error (i.e., $V_i \in \mathcal{Z}$) does not change the relevant mixing matrix $\mathbf{W}^{ME}$. Specifically, in the overall mixing matrix $\mathbf{W}$, cf. Equation (6), the exogenous noise terms associated with non-leaf underlying variables correspond to columns with at least two non-zero entries. This follows because a non-leaf variable influences at least two other variables (its own measurement and its child(ren) in $\mathcal{Z} \cup \mathcal{Y}$). On the other hand, the measurement error terms associated with the non-leaf underlying variables correspond to columns with only one non-zero entry, since the measurement error term only influences one variable (i.e., the corresponding measurement in $U$). Thus, by removing the identity matrix part of $\mathbf{W}$, the remaining part, i.e., $\mathbf{W}^{ME}$ does not change if a non-leaf underlying variable is observed or measured with error.

For a linear SEM-ME, $M'$, without loss of generality, suppose that all non-leaf underlying variables are observed. That is, all `nu-leaf` nodes belong to $\mathcal{Y}$, and `u-leaf` nodes belong to $\mathcal{Z}$. Denote the dimension of $Y$ (i.e., the number of the `nu-leaf` nodes) as $p_n$. Let us first permute the variables in $Y$ in the reversed causal order, and denote the result of this permutation by $Y^*$. Thus,

$$Y^* = \mathbf{P}Y \implies \begin{bmatrix} Z^L \\ Y^* \end{bmatrix} = \begin{bmatrix} \mathbf{D} \\ \mathbf{C}^* \end{bmatrix} Y^* + \begin{bmatrix} 0 \\ N_{Y^*} \end{bmatrix},$$

where $\mathbf{P}$ is a $p_n \times p_n$ permutation matrix, and $N_{Y^*} = \mathbf{P}N_Y$ is the permuted noise vector corresponding to $Y^*$. Further, $\mathbf{C}^* = \mathbf{P}\mathbf{C}\mathbf{P}^\top$ is a row and column permuted version of $\mathbf{C}$, such that $\mathbf{C}^*$ is strictly upper triangular.

Therefore, the mixing matrix $\mathbf{W}^{ME}$ corresponding to $M'$ can be written as

$$\mathbf{W}^{ME}(M') = \begin{bmatrix} \mathbf{I} & \mathbf{0} \\ \mathbf{0} & \mathbf{P}^\top \end{bmatrix} \mathbf{W}_*^{ME} \mathbf{P}, \quad \text{where} \quad \mathbf{W}_*^{ME} = \begin{bmatrix} \mathbf{D}(\mathbf{I} - \mathbf{C}^*) \\ (\mathbf{I} - \mathbf{C}^*) \end{bmatrix}. \tag{10}$$

Next, we derive the mixing matrix of the corresponding SEM-UR under the mapping $\phi$. Suppose the corresponding SEM-UR can be written as

$$H_\phi = N_{H_\phi}, \qquad X_\phi = \mathbf{B}_\phi H_\phi + \mathbf{A}_\phi X_\phi + N_{X_\phi}.$$

Note that under mapping $\phi$, all `u-leaf` nodes in $M'$ are mapped to latent variables in $H_\phi$, and all `nu-leaf` nodes in $M'$ are mapped to observed variables in $X_\phi$. Further, all edges are reversed. This means that there exists a permutation of $H_\phi$, denoted by $H_\phi^*$, such that $Z_i$ is mapped to $H_{\phi_i}^*$ and $Y_i^*$ is mapped to $X_i$. Note that the $X$ variables are arranged according to a certain causal order, which is not necessarily the case for the $Y$ variables. Further, if there is an edge from $Y_i^*$ to $Y_j^*$ in $M'$ with weight $w$, then there exists an edge from $X_i$ to $X_j$ in $\phi(M')$ with weight $w$. Therefore, $\mathbf{A}_\phi = (\mathbf{C}^*)^\top$. Similarly, $\mathbf{B}_\phi \mathbf{P}_H = \mathbf{D}^\top$, where $\mathbf{P}_H$ represents the re-permutation of the latent variables corresponding to the order of the `u-leaf` nodes. The mixing matrix of $\phi(M')$ can be written as

$$\mathbf{W}^{UR}(M) = \left[ \left( \mathbf{I} - (\mathbf{C}^*)^\top \right) \mathbf{D}^\top \mathbf{P}_H^\top \ \left( \mathbf{I} - (\mathbf{C}^*)^\top \right) \right] = \left[ (\mathbf{I} - \mathbf{C}^*)^\top \mathbf{D}^\top \mathbf{P}_H^\top \ (\mathbf{I} - \mathbf{C}^*)^\top \right]. \tag{11}$$

Comparing Equation (10) to Equation (11), we have

$$\mathbf{W}^{ME}(M') = \begin{bmatrix} \mathbf{I} & \mathbf{0} \\ \mathbf{0} & \mathbf{P}^\top \end{bmatrix} \begin{bmatrix} \mathbf{P}_H^\top & \mathbf{0} \\ \mathbf{0} & \mathbf{I} \end{bmatrix} \left( \mathbf{W}^{UR}(M) \right)^\top \mathbf{P}$$

Therefore, there exist column permutations of $\mathbf{W}^{ME}(M')$ and $\mathbf{W}^{UR}(M)$, namely,

$$\tilde{\mathbf{W}}^{ME}(M') = \mathbf{W}^{ME}(M')\mathbf{P}^\top, \qquad \tilde{\mathbf{W}}^{UR}(M) = \mathbf{W}^{UR}(M) \begin{bmatrix} \mathbf{P}_H & \mathbf{0} \\ \mathbf{0} & \mathbf{P} \end{bmatrix}$$

such that $\tilde{\mathbf{W}}^{ME}(M') = (\tilde{\mathbf{W}}^{UR}(M))^\top$.

## C.2 Proof of Theorem 2

In order to prove Theorem 2, we need to prove Propositions 1 and 2, and that the AOG equivalence class is the extent level of identifiability under Assumption 2(a). We begin by proving Proposition 2 (and its counterpart for direct ordered grouping). We formulate these in the following lemma.

**Lemma 1** *Given the mixing matrix $\mathbf{W}^{ME}$ (resp. $\mathbf{W}^{UR}$) and the AOG/DOG of the model, all the coefficients are identified (up to the permutation and scaling of the columns of $\mathbf{B}$ in SEM-UR) if and only if the centers (resp. exogenous noise assignments) of the ancestral/direct ordered groups are identified.*

*Proof.* "only-if direction": If all the coefficients are identified, i.e., matrices $\mathbf{C}, \mathbf{D}$ in Equation (2) (and matrices $\mathbf{A}, \mathbf{B}$ in Equation (4)) are identified, then the structure of the model is identified. Therefore, we can identify ancestral/direct ordered grouping of the model and find the center of each ancestral/direct ordered group.

"if direction": For SEM-ME, given the identified centers in each ordered group, we compute the coefficients of the model from the mixing matrix $\mathbf{W}^{ME}$ as follows. According to the definition of AOG/DOG, only the first $p_n$ ordered groups can include more than one element; recall that $p_n$ denotes the number of `nu-leaf` nodes. Therefore, for any possible selection of the centers in each group, consider the submatrix of $\mathbf{W}^{ME}$ where the rows correspond to the first $p_n$ selected centers. This submatrix can be permuted to a lower-triangular matrix with non-zero diagonal, and maps to $(\mathbf{I} - \mathbf{C})^{-1}$ in Equation (5); recall that $\mathbf{C}$ represents the edges among the first $p_n$ centers (i.e., the `nu-leaf` nodes in the model). That is, $\mathbf{C}$ can be identified by first normalizing the columns of $\mathbf{W}^{ME}$, and then taking the inverse of the submatrix that corresponds to the identified centers. Note that the lower-triangular matrix (after permutation) must have diagonal one. If the diagonal entry has value not equal to one, then this number represents the noise coefficient, and can be removed by column normalization. Matrix $\mathbf{D}$, which represents the edges from `nu-leaf` nodes to `u-leaf` nodes can be deduced from remaining submatrix of $\mathbf{W}^{ME}$ that maps to $\mathbf{D}(\mathbf{I} - \mathbf{C})^{-1}$. Therefore, all coefficients can be identified.

For SEM-UR, given the exogenous noise terms of the identified centers in each ordered group, we use a similar method to compute the coefficients of the model from the mixing matrix $\mathbf{W}^{UR}$. Specifically, $\mathbf{A}$ in Equation (4) can be identified using the submatrix of $\mathbf{W}^{UR}$ where the columns correspond to the last $q$ selected centers ($q$ denotes the number of observed variables). Matrix $\mathbf{B}$ can be deduced from the remaining submatrix of $\mathbf{W}^{UR}$. Note that permuting and normalizing the columns of the submatrix that maps to $(\mathbf{I} - \mathbf{A})^{-1}$ does not affect the the submatrix that maps to $\mathbf{B}(\mathbf{I} - \mathbf{A})^{-1}$. This means that the true permutation and scaling of $\mathbf{B}$ cannot be identified. ∎

In the remainder of the proof of Theorem 2, we focus on linear SEM-ME; the proof for linear SEM-UR follows similarly. The proof of the remaining assertions consists of two parts. In the first part, we show that models in the same AOG equivalence class satisfy Proposition 1 and Assumption 2(a). That is, given the mixing matrix $\mathbf{W}^{ME}$, under different choices of the center in each ancestral ordered group, the alternative model (1) satisfies Assumption 2(a), and (2) preserves the causal order among the ancestral ordered groups. In the second part, we show that any alternative model that has the same mixing matrix as the ground-truth but does not belong to the same AOG equivalence class violates Assumption 2(a). We can see from the proof of Lemma 1 that the model can be identified once all the `nu-leaf` nodes (i.e., variables corresponding to the rows of $\mathbf{C}$) are identified. Therefore, we show that, if at least one of the first $p_n$ ancestral ordered groups in the ground-truth does not include any of the `nu-leaf` nodes in the alternative model, then the alternative model violates Assumption 2(a).

We use $Pa_M(V_i)$ and $An_M(V_i)$ to denote the parent set and ancestor set of variable $V_i$ in model $M$ throughout the remaining of the proof.

**Proof of the first part**:

*Proof of (1)*: Note that Assumption 2(a) (i.e., conventional faithfulness assumption) is equivalent to the following: If $V_i$ is an ancestor of $V_j$, then the support of the row of $\mathbf{W}^{ME}$ corresponding to $V_i$ is a subset of the support of the row of $\mathbf{W}^{ME}$ corresponding to $V_j$. Further, the converse is also true given that $V_i$ is an `nu-leaf` node.

Let $M$ denote the ground-truth, and let $M'$ denote an alternative model that belongs to the same AOG equivalence class as $M$. We first show that if some center nodes in $M$ are swapped with other nodes in their respective ancestral ordered groups, then an added edge(s) to any node $V$ in $M'$ can only originate from variables in $An_M(V)$. To show this, suppose that $V_{i_1}, \cdots, V_{i_m}$ represent some centers of the ancestral ordered groups in $M$, where $m \leq p_n$, and $\{i_1, \cdots, i_m\}$ is consistent with the causal order among the `nu-leaf` nodes in $M$. Now, the centers in $M'$ are changed from $V_{i_1}, \cdots, V_{i_m}$ to $V_{j_1}, \cdots, V_{j_m}$, where $V_{i_k}$ and $V_{j_k}$ are in the same ancestral ordered group, $\forall k \in [m]$.

Consider the structural equations of $V_{i_1}$ (first changed center in $M$) and $V_{j_1}$ (the swapped center in $M'$), and $V_{k_1}$ which is any child of $V_{i_1}$ in $M$ (other than $V_{k_1}$). Since $V_{i_1}$ and $V_{j_1}$ belong to the same ancestral ordered group, if follows by the definition of AOG that parents of $V_{j_1}$, except for $V_{i_1}$, are also ancestors of $V_{i_1}$ in $M$. Therefore the structural equations are:

$$V_{i_1} = \sum_{l:V_l \in Pa_M(V_{i_1})} a_{i_1 l} V_l + N_{V_{i_1}},$$

$$V_{j_1} = a_{j_1 i_1} V_{i_1} + \sum_{l_j:V_{l_j} \in An_M(V_{i_1})} a_{j_1 l_j} V_{l_j},$$

$$V_{k_1} = a_{k_1 i_1} V_{i_1} + \sum_{l_k:V_{l_k} \in Pa_M(V_{k_1})} a_{k_1 l_k} V_{l_k} + N_{V_{k_1}}.$$

Now, we choose $V_{j_1}$ to be the center of the corresponding ordered group in $M'$ (replacing the center $V_{i_1}$ in $M$). Now, in $M'$, $V_{i_1}$ is a child of $V_{j_1}$ and is a `u-leaf` node. Also, the children of $V_{i_1}$ in $M$ are now children of $V_{j_1}$ in $M'$. Therefore, the structural equations of $V_{i_1}$, $V_{j_1}$ and $V_{k_1}$ in $M'$ are

$$V_{j_1} = \sum_{l_j:V_{l_j} \in An_M(V_{i_1})} a_{j_1 l_j} V_{l_j} + \sum_{l:V_l \in Pa_M(V_{i_1})} a_{j_1 i_1} a_{i_1 l} V_l + a_{j_1 i_1} N_{V_{i_1}},$$

$$V_{i_1} = a_{j_1 i_1}^{-1} V_{j_1} + \sum_{l_j:V_{l_j} \in An_M(V_{i_1})} a_{j_1 i_1}^{-1} a_{j_1 l_j} V_{l_j},$$

$$V_{k_1} = a_{k_1 i_1} \left( a_{j_1 i_1}^{-1} V_{j_1} + \sum_{l_j:V_{l_j} \in An_M(V_{i_1})} a_{j_1 i_1}^{-1} a_{j_1 l_j} V_l \right) + \sum_{l_k:V_{l_k} \in Pa_M(V_{k_1})} a_{k_1 l_k} V_{l_k} + N_{V_{k_1}}.$$

Comparing both sets of structural equations, we see that the additional edges to $V_{j_1}$ and $V_{k_1}$ in $M'$ originate from variables in $An_M(V_{i_1})$. Since $V_{i_1}$ is a parent of both, this means that variables in $An_M(V_{i_1})$ are also ancestors of $V_{j_1}$ and $V_{k_1}$. This further implies that the ancestor sets of all variables remain the same after this change, i.e., if $V_i$ is an ancestor of $V_j$ in $M$, then $V_i$ (or the node that replaces $V_i$) is an ancestor of $V_j$ (or the node that replaces $V_j$) in $M'$. By repeating the same procedure, i.e., swapping $V_{i_k}$ with $V_{j_k}$ for $k = 2, 3, \cdots, m$, the resulting model $M'$ will only add edges from variables in $An_M(V)$ to $V$, $\forall V \in \mathcal{Y} \cup \mathcal{Z}$.

Next, we show, by contradiction, that if $M$ satisfies Assumption 2(a), then $M'$ also satisfies Assumption 2(a). Suppose $M'$ violates Assumption 2(a). This means that there exist $V_0$ and $V$ such that $V_0$ is an ancestor of $V$ in $M'$ but the support of $V_0$ (i.e., its corresponding row in $\mathbf{W}^{ME}$) is not a subset of the support of $V$. Note that since $V_0$ is an `nu-leaf` node in $M'$, hence it is a center node of some ancestral ordered group in $M'$. Let $V_0'$ denote the center of the ancestral ordered group of $V_0$ in $M$ (which can be $V_0$ itself). Note that $V_0$ has the

same support as $V_0'$ since they belong to the same ancestral ordered group. Therefore, the support of $V_0'$ is not a subset of the support of $V$. Since $M$ satisfies Assumption 2(a), this means that $V_0'$ is not an ancestor of $V$ in $M$. However, in obtaining $M'$ from $M$ (i.e., by switching some centers in $M$), the edges added to $V$ in $M'$ only come from ancestors of $V$ in $M$. This means that $V_0$ cannot be an ancestor of $V$ in $M'$, which leads to a contradiction. Therefore, $M'$ satisfies Assumption 2(a).

*Proof of (2)*: Given that the alternative model $M'$ satisfies Assumption 2(a), we show that $M'$ preserves the causal order among the ancestral ordered groups. We show this by contradiction. Suppose that a variable $V_0$ in a later group causes another variable $V$ in an earlier group in $M'$; according to the causal order among the groups in $M$. Note that $V_0$ is an `nu-leaf` node in $M'$ since it causes $V$, and hence it is the center of some ancestral ordered group in $M'$. Let $V_0'$ be the center of the corresponding group in $M$, with an associated exogenous noise term denoted by $N_{V_0'}$. Since $V_0$ and $V_0'$ belong to the same ancestral ordered group, $N_{V_0'}$ must belong to the support of $V_0$. However, since $V$ is in an earlier group than $V_0$ in $M'$, $N_{V_0'}$ is not in the support of $V$. Therefore the support of $V_0$ is not a subset of the support of $V$, and hence, $M'$ violates Assumption 2(a).

**Proof of the second part**:

If the alternative model $M'$ does not belong to the AOG equivalence class of the ground-truth $M$, then at least one of the first $p_n$ ancestral ordered groups in $M$ does not include any of the `nu-leaf` nodes in $M'$. Denote the center of this ordered group as $V_i$, and let $N_{V_i}$ denote its exogenous noise. Since $V_i$ is a `u-leaf` node in $M'$, there is at least one parent of $V_i$ in $M'$ that also includes $N_{V_i}$ (i.e., is affected by $N_{V_i}$ directly or indirectly). Denote this variable as $V_j$. Since $V_j$ includes $N_{V_i}$, it must be a descendant of $V_i$ in $M$. As $M$ satisfies Assumption 2(a), this means that the support of $V_j$ (in $\mathbf{W}^{ME}$) is a superset of the support of $V_i$. Further, since the ancestral ordered group of $V_i$ does not include any `nu-leaf` nodes in $M'$, $V_j$ does not belong to the ancestral ordered group of $V_i$. This means that the support of $V_j$ (in $\mathbf{W}^{ME}$) is not the same as the support of $V_i$. Therefore, the support of $V_j$ (in $\mathbf{W}^{ME}$) is a strict superset of the support of $V_i$. Since $V_j$ is a parent of $V_i$ in $M'$, this violates Assumption 2(a) on the alternative model $M'$.

**Conclusion**: We showed that by under different choices of the centers in the ancestral ordered groups, we can always find a model $M'$ that: (1) has the same mixing matrix; (2) satisfies Assumption 2(a); (3) preserves the causal order among ancestral ordered groups. Thus, we cannot distinguish $M'$ from the ground-truth under Assumption 2(a) alone. Further, we show that any model that have the same mixing matrix but does not belong to the AOG equivalence class does violate Assumption 2(a) and can be distinguished from the ground-truth $M$. Therefore the extent level of identifiability can be characterized by AOG equivalence class.

## C.3    Proof of Theorem 3

To prove Theorem 3, we need to prove Propositions 3 and 4, and that the DOG equivalence class is the extent level of identifiability under Assumption 2. The proof can be divided into two parts. We first show the necessity part, that is, models in the same DOG equivalence class as the ground-truth satisfy Assumption 2 and Proposition 3, hence cannot be distinguished from the ground-truth. We then show the sufficiency part, i.e., any model that has the same mixing matrix but does not belong to the DOG equivalence class of the ground-truth violates Assumption 2.

### C.3.1    Proof of necessity

We have shown in Lemma 1 that models in the same DOG equivalence class can be identified by the choice of centers of the direct ordered groups. In the following, given the ground-truth model $M$ and a certain choice of the centers that corresponds to an alternative model $M'$, where $M'$ belongs to the same DOG equivalence class as $M$, we show that in the alternative model: (1) No edges are added; (2) No edges are removed; (3) Faithfulness is preserved. Therefore we cannot distinguish the alternative model from the ground truth.

Suppose that $V_{i_1}, \cdots, V_{i_m}$ represent some centers of the direct ordered groups in the ground-truth $M$, where $m \leq p_n$, and $\{i_1, \cdots, i_m\}$ is consistent with the causal order among the `nu-leaf` nodes in $M$. Now, the centers in $M'$ are changed from $V_{i_1}, \cdots, V_{i_m}$ to $V_{j_1}, \cdots, V_{j_m}$, where $V_{i_k}$ and $V_{j_k}$ are in the same direct ordered group, $\forall k \in [m]$. It follows by the definition of DOG that the edge $V_{i_k} \to V_{j_k}$ violates Condition 1, $\forall k \in [m]$.

Note that according to Condition 1, we have the following two properties: (1) For each `nu-leaf` node $V$ and each `u-leaf` node $V'$ that belongs to the same direct ordered group as $V$, $Pa(V') \setminus \{V\} \subseteq Pa(V)$. (2) All `u-leaf` nodes that belong to the same direct ordered group have the same parent set; otherwise, Condition 1 will be satisfied and the `u-leaf` nodes cannot belong to the same direct ordered group.

Lastly, for any model $M$, any set of variables $\mathcal{V}$ and variable $V_j$, let $\text{TE}_M(\mathcal{V}, V_j)$ denote the vector of total causal effects from variables in $\mathcal{V}$ to $V_j$ in model $M$. Note that the total effect from a variable $V_i$ to $V_j$ in $M_0$ is equal to the entry in $\mathbf{W}^{ME}$ with row corresponding to $V_j$ and column corresponding to $N_{V_i}$. Since $M$ and $M'$ has the

same mixing matrix, if $N_{V_i}$ is the exogenous noise term of another variable $V_i'$ in $M'$, then the total effect from a variable $V_i$ to $V_j$ in $M$ is equal to the total effect from $V_i'$ to $V_j$ in $M'$, i.e., $\text{TE}_{M_0}(\{V_i\}, V_j) = \text{TE}_{M'}(\{V_i'\}, V_j)$.

**Proof of (1)**: Consider a set of models $M_0, M_1, \cdots, M_m$, where $M_0 = M$, and $M_k$ changes the centers $V_{i_1} \cdots, V_{i_k}$ in $M_0$ to $V_{j_1} \cdots, V_{j_k}$; $k \in [m]$. In the following, we show that for all $k \in [m]$, obtaining $M_k$ from $M_0$ as described above does not add extra edges in $M_k$.

We start with $k = 1$. The structural equations of $V_{i_1}$ and $V_{j_1}$ in $M_0$ are:

$$V_{i_1} = \sum_{l:V_l \in Pa_{M_0}(V_{i_1})} a_{i_1 l} V_l + N_{V_{i_1}}, \tag{12}$$

$$V_{j_1} = \sum_{l_j:V_{l_j} \in Pa_{M_0}(V_{j_1}) \setminus \{V_{i_1}\}} a_{j_1 l_j} V_{l_j} + a_{j_1 i_1} V_{i_1}, \tag{13}$$

Further, the structural equation of any child $V_{k_1}$ of $V_{i_1}$ in $M_0$ is:

$$V_{k_1} = \sum_{l_k:V_{l_k} \in Pa_{M_0}(V_{k_1}) \setminus \{V_{i_1}\}} a_{k_1 l_k} V_{l_k} + a_{k_1 i_1} V_{i_1} + N_{V_{k_1}}. \tag{14}$$

Note that $N_{V_{k_1}} = 0$ if and only if $V_{k_1}$ is a `u-leaf` node.

In $M_1$, since we only swap $V_{i_1}$ and $V_{j_1}$, the structural equations of all variables in $\mathcal{Z} \cup \mathcal{Y} \setminus \{V_{i_1}, V_{j_1}\} \cup Ch_{M_0}(V_{i_1})$ remain the same as those in $M_0$. From (12) and (13), the structural equation of $V_{j_1}$ in $M_1$:

$$V_{j_1} = \sum_{l_j:V_{l_j} \in Pa_{M_0}(V_{j_1}) \setminus \{V_{i_1}\}} a_{j_1 l_j} V_{l_j} + a_{j_1 i_1} \left( \sum_{l:V_l \in Pa_{M_0}(V_{i_1})} a_{i_1 l} V_l + N_{V_{i_1}} \right)$$

$$= \sum_{l:V_l \in Pa_{M_0}(V_{i_1})} a_{j_1 i_1} a_{i_1 l} V_l + \sum_{l_j:V_{l_j} \in Pa_{M_0}(V_{j_1}) \setminus \{V_{i_1}\}} a_{j_1 l_j} V_{l_j} + a_{j_1 i_1} N_{V_{i_1}}.$$

Since the edge $V_{i_1} \to V_{j_1}$ violates Condition 1(a), $Pa_{M_0}(V_{j_1}) \setminus \{V_{i_1}\} \subseteq Pa_{M_0}(V_{i_1})$. By defining $a'_{j_1 l} \triangleq a_{j_1 i_1} a_{i_1 l} + a_{j_1 l} \mathbf{1}[V_l \in Pa_{M_0}(V_{j_1})]$ for all $V_l \in Pa_{M_0}(V_{i_1})$, we can re-write $V_{j_1}$ as

$$V_{j_1} = \sum_{V_l \in Pa_{M_0}(V_{i_1})} a'_{j_1 l} V_l + a_{j_1 i_1} N_{V_{i_1}}. \tag{15}$$

Next, from (13), the structural equation of $V_{i_1}$ in $M_1$ is:

$$V_{i_1} = \frac{1}{a_{j_1 i_1}} V_{j_1} - \sum_{l_j:V_{l_j} \in Pa_{M_0}(V_{j_1}) \setminus \{V_{i_1}\}} \frac{a_{j_1 l_j}}{a_{j_1 i_1}} V_{l_j}. \tag{16}$$

Lastly, by substituting $V_{i_1}$ in (14) by (16), the structural equation of $V_{k_1}$ in $M_1$ is:

$$V_{k_1} = \sum_{l_k:V_{l_k} \in Pa_{M_0}(V_{k_1}) \setminus \{V_{i_1}\}} a_{k_1 l_k} V_{l_k} + a_{k_1 i_1} \left( \frac{1}{a_{j_1 i_1}} V_{j_1} - \sum_{l_j:V_{l_j} \in Pa_{M_0}(V_{j_1}) \setminus \{V_{i_1}\}} \frac{a_{j_1 l_j}}{a_{j_1 i_1}} V_{l_j} \right) + N_{V_{k_1}}.$$

Since $V_{i_1} \to V_{j_1}$ violates Condition 1(b), $Pa_{M_0}(V_{j_1}) \subseteq Pa_{M_0}(V_{k_1})$. Therefore, by defining $a'_{k_1 l_k} \triangleq a_{k_1 l_k} - \frac{a_{k_1 i_1} a_{j_1 l_k}}{a_{j_1 i_1}} \mathbf{1}[V_{l_k} \in Pa_{M_0}(V_{j_1})]$, we can re-write $V_{k_1}$ as

$$V_{k_1} = \sum_{l_k:V_{l_k} \in Pa_{M_0}(V_{k_1}) \setminus \{V_{i_1}\}} a'_{k_1 l_k} V_{l_k} + \frac{a_{k_1 i_1}}{a_{j_1 i_1}} V_{j_1} + N_{V_{k_1}}. \tag{17}$$

Recall that in $M_1$, the structural equations of all underlying variables in $\mathcal{Z} \cup \mathcal{Y} \setminus \{V_{i_1}, V_{j_1}\} \cup Ch_{M_0}(V_{i_1})$ are the same as $M_0$. Further, the structural equations of $V_{i_1}$, $V_{j_1}$ and any $V_{k_1} \in Ch_{M_0}(V_{i_1})$ in (12)-(14) are replaced by (15)-(17). Thus, we have $Pa_{M_1}(V_{j_1}) = Pa_{M_0}(V_{i_1})$, $Pa_{M_1}(V_{i_1}) = \{V_{j_1}\} \cup Pa_{M_0}(V_{j_1}) \setminus \{V_{i_1}\}$, and $Pa_{M_1}(V_{k_1}) = \{V_{j_1}\} \cup Pa_{M_0}(V_{k_1}) \setminus \{V_{i_1}\}$. Therefore no edge will be added in $M_1$, and $V_{i_g} \to V_{j_g}$ still violates Condition 1 in $M_1$ for all $g = [2 : m]$.

Note that $Pa_{M_1}(V_{j_1})$ (resp. $Pa_{M_1}(V_{k_1})$) may be a subset of $Pa_{M_0}(V_{i_1})$ (resp. $\{V_{j_1}\} \cup Pa_{M_0}(V_{k_1}) \setminus \{V_{i_1}\}$) due to certain parameter cancellation in $a'_{j_1 l}$ (resp. $a'_{k_1 l_k}$). However, it does not affect our conclusion here as we show that no edges are added in $M_1$. We will show in Part (2) that if this parameter cancellation happens (at any point of the procedure of swapping centers), then the model violates faithfulness assumption.

Next, consider $k = 2$. Repeating the same procedure of swapping centers, the structural equations of $M_2$ can be obtained from $M_1$. Similarly, we have $Pa_{M_2}(V_{j_2}) = Pa_{M_1}(V_{i_2})$, $Pa_{M_2}(V_{i_2}) = \{V_{j_2}\} \cup Pa_{M_1}(V_{j_2}) \setminus \{V_{i_2}\}$,

and $Pa_{M_2}(V_{k_2}) = \{V_{j_2}\} \cup Pa_{M_1}(V_{k_2}) \setminus \{V_{i_2}\}$. Therefore, no edge will be added when obtaining $M_2$ from $M_0$.

By repeating the same procedure for $k = 3, \cdots, m$, we conclude that obtaining the final model $M_m$ from $M_0$ does not add additional edges in $M_m$ compared to $M_0$.

**Proof of (2)**: Following the procedure of swapping centers described in Part (1), we show that if any edge from $M_0$ is removed in $M_m$, then the ground-truth model $M_0$ violates Assumption 2. Suppose the edge from $V_0$ to $V$ in $M_0$ is removed in $M_m$. Here "edge removal" means that there is an edge from $V_0$ to $V$ in $M_0$, but there is no edge from $V_0$ (or the node that replaces $V_0$) to $V$ (or the node that replaces $V$) in $M_m$. Note that $V_0$ is an `nu-leaf` node in $M_0$, which is not necessarily the case for $V$. Hence we consider each of the following cases:

(i) $V$ is not swapped with any other nodes, i.e., $V \notin \{V_{i_1}, \cdots, V_{i_m}, V_{j_1} \cdots, V_{j_m}\}$. This means that $V_0$ is a parent of $V$ in $M_0$, but $V_0$ (or the node that replaces $V_0$) is not a parent of $V$ in $M_m$.

(ii) $V = V_{i_k}$ for some $k \in [m]$. This means that $V_0$ is a parent of a center node $V_{i_k}$ in $M_0$, but $V_0$ (or the node that replaces $V_0$) is not a parent of the center node $V_{j_k}$ in $M_m$.

(iii) $V = V_{j_k}$ for some $k \in [m]$. This means that $V_0$ is a parent of a non-center node $V_{j_k}$ in $M_0$, but $V_0$ (or the node that replaces $V_0$) is not a parent of the non-center node $V_{i_k}$ in $M_m$.

*Case (i).* Suppose $V$ is not swapped with any other nodes. Since the edge from $V_0$ to $V$ is removed in $M_m$, $V$ has at least one less parent in $M_m$ than in $M_0$. This means that the total effects of $An_{M_m}(V)$ on $V$, i.e., $\text{TE}_{M_m}(An_{M_m}(V), V)$, can be written as a linear combination of the total effects of $An_{M_m}(V)$ on parents of $V$ in $M_m$, i.e., vectors in $\{\text{TE}_{M_m}(An_{M_m}(V), V') : V' \in Pa_{M_m}(V)\}$. Further, following the procedure of swapping centers, all parents of $V$ in $M_m$ are possible parents of $V$ in $M_0$. Note that $\text{TE}_{M_m}(An_{M_m}(V), V) = \text{TE}_{M_0}(An_{M_0}(V), V)$, and for any possible parents $V'$ of $V$ in $M_0$, $\text{TE}_{M_m}(An_{M_m}(V), V') = \text{TE}_{M_0}(An_{M_0}(V), V')$. Therefore, $V$ violates Assumption 2(b1) on $M_0$.

*Case (ii).* Suppose $V = V_{i_k}$ for some $k \in [m]$. Since the edge from $V_0$ to $V_{j_k}$ is removed in $M_m$, $V_{j_k}$ has at least one less parent in $M_m$ than $V_{i_k}$ has in $M_0$. This means that $\text{TE}_{M_m}(An_{M_m}(V_{j_k}), V_{j_k})$ can be written as a linear combination of at most $|Pa_{M_0}(V_{i_k})| - 1$ vectors in $\{\text{TE}_{M_m}(An_{M_m}(V_{j_k}), V') : V' \in Pa_{M_m}(V_{j_k})\}$. Note that $|Pa_{M_0}(V_{i_k}) \cup Pa_{M_0}(V_{j_k})| - 1 = |Pa_{M_0}(V_{i_k})|$ since $Pa_{M_0}(V_{j_k}) \setminus \{V_{i_k}\}$ is a subset of $Pa_{M_0}(V_{i_k})$. Further, all parents of $V_{j_k}$ in $M_m$ are possible parents of $V_{i_k}$ in $M_0$. Note that $\text{TE}_{M_0}(An_{M_0}(V_{j_k}) \setminus \{V_{i_k}\}, V_{j_k}) = \text{TE}_{M_m}(An_{M_m}(V_{j_k}), V_{j_k})$, and for any possible parents $V'$ of $V_{i_k}$ in $M_0$, $\text{TE}_{M_0}(An_{M_0}(V_{j_k}) \setminus \{V_{i_k}\}, V') = \text{TE}_{M_m}(An_{M_m}(V_{j_k}), V')$. Therefore, $V_{j_k}$ violates Assumption 2(b2) on $M_0$.

*Case (iii).* Suppose $V = V_{j_k}$ for some $k \in [m]$. Note that following the procedure of swapping centers, for any $g > k$, replacing $V_{i_g}$ by $V_{j_g}$ does not change the structural equations of $V_{i_k}$ and $V_{j_k}$. Therefore, if $V_0$ (or the node that replaces $V_0$) is not a parent of $V_{i_k}$ in $M_m$, then it is not a parent of $V_{i_k}$ in $M_k$. Further, according to Equations (13) and (16), when replacing $V_{j_k}$ by $V_{i_k}$, the parent set of $V_{i_k}$ in $M_k$, excluding $V_{j_k}$, is exactly the same as the parent set of $V_{j_k}$ before the swapping (i.e., in $M_{k-1}$) excluding $V_{i_k}$. Therefore, $V_0$ (or the node that replaces $V_0$) is not a parent of $V_{j_k}$ in $M_{k-1}$, while $V_0$ is a parent of a $V_{j_k}$ in $M_0$. Recall that $V_{j_k}$ is not swapped with any other nodes from $M_0$ to $M_{k-1}$. Therefore this reduces to the Case (i) above, which indicates that $V_{j_k}$ violates Assumption 2(b1) in $M_0$.

In conclusion, if the ground-truth satisfies Assumption 2, then the procedure of swapping centers described in Part (1) does not remove any edge from $M_0$.

**Proof of (3)**: From Parts (1) and (2), we conclude that the structure of the new model $M_m$ is isomorphic to $M_0$. Specifically, we can construct a mapping $\sigma$ from the nodes in $M_0$ to the nodes in $M_m$, where

$$\sigma(V) = V, \quad V \in \mathcal{Z} \cup \mathcal{Y} \setminus \{V_{i_1}, \cdots, V_{i_m}, V_{j_1}, \cdots, V_{j_m}\};$$
$$\sigma(V_{i_k}) = V_{j_k}, \quad \sigma(V_{j_k}) = V_{i_k}, \quad \forall k \in [m].$$

Under this mapping, there is an edge from node $V'$ to $V''$ in $M_0$ if and only if this is an edge from $\sigma(V')$ to $\sigma(V'')$ in $M_m$, for all $V', V'' \in \mathcal{Z} \cup \mathcal{Y}$. Note that this mapping only switches pairs of nodes that belong to the same direct ordered group. This means that $M_m$ has the same edges across groups, which completes the proof of Proposition 3.

In the following, we show that if $M_0$ satisfies Assumption 2, then $M_m$ satisfies Assumption 2. Note that since the DOG is a more refined partition of AOG, according to the proof of Theorem 2, Assumption 2(a) is satisfied. Therefore we only need to verify Assumption 2(b). For each variable $V$, we consider each of the three following cases:

(i) $V \notin \{V_{i_1}, \cdots, V_{i_m}, V_{j_1} \cdots, V_{j_m}\}$, i.e., $V$ is not changed from $M_0$ to $M_m$.

(ii) $V = V_{i_k}$ for some $k \in [m]$, i.e., $V$ is a center node in $M_0$ and is changed to a non-center node in $M_m$.

(iii) $V = V_{j_k}$ for some $k \in [m]$, i.e., $V$ is a non-center node in $M_0$ and is changed to a center node in $M_m$.

*Case (i).* Suppose $V \notin \{V_{i_1}, \cdots, V_{i_m}, V_{j_1} \cdots, V_{j_m}\}$. Note that any possible parent of $V$ in $M_0$ is also a possible parent of $V$ in $M_m$, and for all $V' \in$ {Possible parents of $V$ in $M_m\} \cup \{V\}$, $\text{TE}_{M_0}(An_{M_0}(V), V') = \text{TE}_{M_m}(An_{M_m}(V), V')$. Further, if the parent set of any u-leaf node $V'$ in $M_0$ is a subset of $Pa_{M_0}(V)$, then the parent set of any u-leaf node in this direct ordered group in $M_m$ is a subset of $Pa_{M_m}(V)$. Therefore Assumption 2*(b1)* holds for $V$ on $M_m$.

Similarly, for Assumption 2*(b2)*, for each parent $V_0$ of u-leaf node $V$ in $M_m$, denote $V'_0$ as the center of the direct ordered group of $V_0$ in $M_m$. Then any possible parent of $V'_0$ in $M_0$ is also a possible parent of $V_0$ in $M_m$, and for all $V' \in$ {Possible parents of $V_0$ in $M_m\} \cup \{V\}$, $\text{TE}_{M_0}(An_{M_0}(V) \setminus \{V'_0\}, V') = \text{TE}_{M_m}(An_{M_0}(V) \setminus \{V_0\}, V')$. Therefore Assumption 2*(b2)* holds for $V$ on $M_m$.

*Case (ii).* Suppose $V = V_{i_k}$ for some $k \in [m]$. Since $V_{i_k}$ is a nu-leaf node in $M_0$, $\text{TE}_{M_0}(An_{M_0}(V_{i_k}), V_{i_k}) = \text{TE}_{M_m}(An_{M_m}(V_{i_k}) \setminus \{V_{j_k}\}, V_{i_k})$. Further, any possible parent of $V_{i_k}$ in $M_0$ is a possible parent of $V_{j_k}$ in $M_m$, and $|Pa_{M_0}(V_{i_k})| = |Pa_{M_m}(V_{j_k})| = |Pa_{M_m}(V_{j_k}) \cup Pa_{M_m}(V_{i_k}) \setminus \{V_{j_k}\}|$. Besides, if the parent set of any u-leaf node $V'$ in $M_0$ is a subset of $Pa_{M_0}(V_{i_k})$, then the parent set of any u-leaf node in this direct ordered group in $M_m$ is also a subset of $Pa_{M_0}(V_{j_k})$. Therefore, since Assumption 2*(b1)* holds for $V_{i_k}$ on $M_0$, Assumption 2*(b2)* holds for $V_{i_k}$ on $M_m$.

As for Assumption 2*(b1)*, note that the set of possible parents of $V_{i_k}$ in $M_m$ includes all other variable in the same ancestral ordered group as $V_{i_k}$ in $M_0$, and is not the same as the set of possible parents of $V_{i_k}$ in $M_0$. If $\text{TE}_{M_m}(An_{M_m}(V_{i_k}), V_{i_k})$ can be written as a linear combination of $k \le |Pa_{M_m}(V_{i_k})|$ vectors from the set $\{\text{TE}_{M_m}(An_{M_m}(V_{i_k}), V') : V' \text{ is a possible parent of } V_{i_k} \text{ in } M_0\}$, denote the set of the possible parents that corresponds to these vectors as $\mathcal{K}$. Then at least one of the variables in $\mathcal{K}$ belongs to the ancestral ordered group of $V_{i_k}$ in $M_0$. Denote this variable as $V$. This means that $\text{TE}_{M_m}(An_{M_m}(V_{i_k}), V)$ can be written as a linear combination of $k$ vectors which includes $\text{TE}_{M_m}(An_{M_m}(V_{i_k}), V_{i_k})$ and $\{\text{TE}_{M_m}(An_{M_m}(V_{i_k}), V') : V' \in \mathcal{K} \setminus \{V\}\}$. Since $V$ is a u-leaf node in $M_0$, any variable in $\mathcal{K} \setminus \{V\}$ is a possible parent of $V$ in $M_0$, and $\text{TE}_{M_m}(An_{M_m}(V_{i_k}), V') = \text{TE}_{M_0}(An_{M_0}(V), V')$ for all $V' \in \mathcal{K} \cup \{V_{i_k}\}$. This means that that $\text{TE}_{M_0}(An_{M_0}(V), V)$ can be written as a linear combination of $k$ vectors which includes $\text{TE}_{M_0}(An_{M_0}(V), V_{i_k})$ and $\{\text{TE}_{M_0}(An_{M_0}(V), V') : V' \in \mathcal{K} \setminus \{V\}\}$. Besides, according to our condition, $|Pa_{M_m}(V_{i_k})| \le |Pa_{M_0}(V)|$ (otherwise $V_{i_k}$ does not belong to the same direct ordered group as $V_{j_k}$). Therefore, since Assumption 2*(b1)* holds for $V$ on $M_0$, $|Pa_{M_0}(V)| = |Pa_{M_m}(V_{i_k})| = k$, and the parent set of the variables in $\mathcal{K}$ that are u-leaf nodes in $M_0$ must be a subset of $Pa_{M_0}(V)$. This implies that parent set of the variables in $\mathcal{K}$ that are u-leaf nodes in $M_m$ must be a subset of $Pa_{M_m}(V_{i_k})$. Therefore, Assumption 2*(b1)* holds for $V_{i_k}$ on $M_m$.

*Case (iii).* Suppose $V = V_{j_k}$ for some $k \in [m]$. Since $V_{j_k}$ is a u-leaf node in $M_0$, $\text{TE}_{M_m}(An_{M_m}(V_{j_k}), V_{j_k}) = \text{TE}_{M_0}(An_{M_0}(V_{j_k}) \setminus \{V_{i_k}\}, V_{j_k})$. Further, any possible parent of $V_{i_k}$ in $M_0$ is a possible parent of $V_{j_k}$ in $M_m$, and $|Pa_{M_m}(V_{j_k})| = |Pa_{M_0}(V_{i_k})|$. Besides, if the parent set of any u-leaf node $V'$ in $M_0$ is a subset of $Pa_{M_0}(V_{i_k})$, then the parent set of any u-leaf node in this direct ordered group in $M_m$ is also a subset of $Pa_{M_m}(V_{j_k})$. Therefore, since Assumption 2*(b2)* holds for $V_{j_k}$ on $M_0$, Assumption 2*(b1)* holds for $V_{j_k}$ on $M_m$.

To sum up, Assumption 2*(b)* holds for every variable on $M_m$. Hence $M_m$ satisfies Assumption 2.

**Conclusion**: We showed that by replacing some centers in the ground-truth $M_0$ by some other nodes in the same direct ordered group, we can get an alternative model $M_m$ that (1) has the same mixing matrix, (2) has an isomorphic graph structure to $M_0$, and (3) satisfies Assumption 2. Therefore we cannot distinguish $M_0$ from any other model in the DOG equivalence class.

### C.3.2   Proof of sufficiency

To prove sufficiency, we show that any model that has the same mixing matrix as the ground-truth but does not belong to the same DOG equivalence class violates Assumption 2. Recall that Theorem 2 implies that any model with the same mixing matrix as the ground-truth, but does not belong to its AOG equivalence class, violates Assumption 2*(a)*. Consequently, it remains to prove that any model that belongs to the AOG equivalence class of the ground-truth, but does not belong to its DOG equivalence class, violates Assumption 2*(b)*. Recall from Proposition 2 that models in the same AOG equivalence class can be identified by the choice of the centers in each ancestral ordered group. Therefore, we show that, given the ground-truth and the choice of the centers, where at least one selected center does not belong to the direct ordered group of the u-leaf node in the ground-truth: (1) at least one edge is added to the ground-truth; (2) no edge in the ground-truth is removed, and at least one added edge is not removed; (3) the alternative model violates Assumption 2*(b)*. Therefore Proposition 4 holds.

In the following, we refer to the centers as the centers of the ancestral ordered groups, unless otherwise stated.

Suppose that $V_{i_1}, \cdots, V_{i_m}$ represent some centers of the ancestral ordered groups in the ground-truth $M$, where $m \le p_n$, and $\{i_1, \cdots, i_m\}$ is consistent with the causal order among the nu-leaf nodes in $M$. Now, the centers in $M'$ are changed from $V_{i_1}, \cdots, V_{i_m}$ to $V_{j_1}, \cdots, V_{j_m}$, where $V_{i_k}$ and $V_{j_k}$ belong to the same ancestral

ordered group, $\forall k \in [m]$. Further, at least one pair of $(V_{i_k}, V_{j_k})$, $k \in [m]$ does not belong to the same direct ordered group.

**Proof of (1)**: Consider the same set of models $M_0, M_1, \cdots, M_m$ as in the necessity proof, where $M_0 = M$, and $M_k$ changes the centers $V_{i_1} \cdots, V_{i_k}$ in $M_0$ to $V_{j_1} \cdots, V_{j_k}$; $k \in [m]$. In the following, we show that if at least one pair of $(V_{i_g}, V_{j_g})$, $g \in [k]$ does not belong to the direct ordered group, then obtaining $M_k$ from $M_0$ as described above adds at least one extra edge.

We start from $k = 1$. We have shown that if $V_{j_1}$ and $V_{i_1}$ are in the same direct ordered group, then obtaining $M_1$ from $M_0$ does not add any edge to the model. Therefore we only focus on the case when $V_{j_1}$ and $V_{i_1}$ are in different direct ordered groups. The true structural equations of $V_{i_1}$ and $V_{j_1}$ and $V_{k_1} \in Ch_{M_0}(V_{i_1})$ in $M_0$ are the same as Equations (12)-(14) so we will not repeat them here. The structural equation of $V_{j_1}$ in $M_1$ is:

$$V_{j_1} = \sum_{l : V_l \in Pa_{M_0}(V_{i_1})} a_{j_1 i_1} a_{i_1 l} V_l + \sum_{l_j : V_{l_j} \in Pa_{M_0}(V_{j_1}) \setminus \{V_{i_1}\}} a_{j_1 l_j} V_{l_j} + a_{j_1 i_1} N_{V_{i_1}}.$$

Since the edge $V_{i_1} \to V_{j_1}$ satisfies Condition 1, $Pa_{M_0}(V_{j_1}) \setminus \{V_{i_1}\}$ may not be a subset of $Pa_{M_0}(V_{i_1})$. Therefore, by defining $a'_{j_1 l} \triangleq a_{j_1 i_1} a_{i_1 l} + a_{j_1 l} \mathbf{1}[V_l \in Pa_{M_0}(V_{j_1})]$ for all $V_l \in Pa_{M_0}(V_{i_1})$, and $\mathcal{S}_{j_1} = Pa_{M_0}(V_{j_1}) \setminus (\{V_{i_1}\} \cup Pa_{M_0}(V_{i_1}))$, the above equation can be written as

$$V_{j_1} = \sum_{V_l \in Pa_{M_0}(V_{i_1})} a'_{j_1 l} V_l + \sum_{l_j : V_{l_j} \in \mathcal{S}_{j_1}} a_{j_1 l_j} V_{l_j} + a_{j_1 i_1} N_{V_{i_1}}. \tag{18}$$

The new structural equation of $i_1$ is the same as Equation (16), since no parameter calculation is required:

$$V_{i_1} = \frac{1}{a_{j_1 i_1}} V_{j_1} - \sum_{l_j : V_{l_j} \in Pa_{M_0}(V_{j_1}) \setminus \{V_{i_1}\}} \frac{a_{j_1 l_j}}{a_{j_1 i_1}} V_{l_j}. \tag{19}$$

As for $V_{k_1}$, using the same analysis as in Equation (18), define $a'_{k_1 l_k} \triangleq a_{k_1 l_k} - \frac{a_{k_1 i_1} a_{j_1 l_k}}{a_{j_1 i_1}} \mathbf{1}[V_{l_k} \in Pa_{M_0}(V_{j_1})]$, and $\mathcal{S}_{k_1} = Pa_{M_0}(V_{j_1}) \setminus Pa_{M_0}(V_{k_1})$. Then structural equation of $V_{k_1}$ is:

$$V_{k_1} = \sum_{l_k : V_{l_k} \in Pa_{M_0}(V_{k_1}) \setminus \{V_{i_1}\}} a'_{k_1 l_k} V_{l_k} + \sum_{l_j : V_{l_j} \in \mathcal{S}_{k_1}} \frac{a_{j_1 l_j}}{a_{j_1 i_1}} V_{l_j} + \frac{a_{k_1 i_1}}{a_{j_1 i_1}} V_{j_1} + N_{V_{k_1}}. \tag{20}$$

Recall that in $M_1$, the structural equations of all underlying variables in $\mathcal{Z} \cup \mathcal{Y} \setminus \{V_{i_1}, V_{j_1}\} \cup Ch_{M_0}(V_{i_1})$ are the same as $M_0$, and the structural equations of $V_{i_1}, V_{i_1}$ and any $V_{k_1} \in Ch_{M_0}(V_{i_1})$ in (12)-(14) are replaced by (18)-(20). Since the edge $V_{i_1} \to V_{j_1}$ satisfies Condition 1, at least one of the sets in $\{\mathcal{S}_{j_1}\} \cup \{\mathcal{S}_{k_1} : k \in Ch_{M_0}(V_{i_1})\}$ must be non-empty. Therefore, compared to $M_0$, there is at least one edge added in $M_1$.

Next consider $k = 2$. Similarly, if $V_{i_2}$ and $V_{j_2}$ are in the same direct ordered group, then no edges are added to $M_2$. If $V_{i_2}$ and $V_{j_2}$ belong to different direct ordered groups, then there is at least one edge added to $M_2$ (compared with $M_0$). Note that this edge may be the same as the edges added to $M_1$. However, we will show in Part (2) that at least one of the the added edges after swapping both centers does not cancel out each other. Therefore it is still an added edge compared with $M_0$.

By repeating the same procedure for $k = 3, \cdots, m$, we conclude that, since at least one of the new centers does not belong the same direct ordered group, at least one more edge is added in $M_m$ compared to $M_0$.

**Proof of (2)**: We first show that if $M_m$ removes any edge from $M_0$, then the ground-truth model $M_0$ violates Assumption 2. The proof is the same as the proof of Part (2) in the necessity proof: We only used the fact that all variables in the same direct ordered group have the same support, which also holds for ancestral ordered group. Therefore if any edge in $M_0$ is removed, then $M_0$ violates Assumption 2*(b)*.

Next, we show that there is at least one added edge that cannot be canceled out. We prove by contradiction, i.e, if all added edges cancel out each other in $M_m$, then $M_0$ violates Assumption 2*(b)*. Suppose the edge from $V_0$ to $V$ is is firstly added in $M_{k_0}$ for some $k_0 \in [m]$ and is removed in $M_m$. Note that $V_0$ is a center node in $M_{k_0}$, but the position of $V_0$ might be changed from $M_0$ to $M_{k_0-1}$. Denote $V'_0$ as the center of the ancestral ordered group of $V_0$ in $M_0$ (which might be $V_0$ itself). Recall from Part (1) that if an edge from $V_0$ to $V$ is added, then there must be a causal path from $V'_0$ to $V$ in $M_0$. We further assume that no edges are added from any ancestor of $V'_0$ in $M_0$ to other nodes, and for any variable $\tilde{V}$ on the causal path from $V'_0$ to $V$, no edges are added from $V'_0$ to $\tilde{V}$. This can be satisfied by selecting $V'_0$ with the smallest index among all added edges following the causal order, and selecting $V$ that belongs to the ancestral ordered group with the smallest index among all added edges from $V_0$. We consider each of the three following cases:

(i) $V$ is not swapped with any other nodes, i.e., $V \notin \{V_{i_1}, \cdots, V_{i_m}, V_{j_1} \cdots, V_{j_m}\}$. This means that the edge from $V_0$ to $V$ is added in $M_{k_0}$, but $V_0$ is not a parent of $V$ in $M_m$.

(ii) $V = V_{i_k}$ for some $k \geq k_0$. This means that the edge from $V_0$ to $V_{i_k}$ is added in $M_{k_0}$, but $V_0$ is not a parent of $V_{j_k}$ in $M_m$.

(iii) $V = V_{j_k}$ for some $k \geq k_0$. This means that the edge from $V_0$ to $V_{j_k}$ is added in $M_{k_0}$, but $V_0$ is not a parent of $V_{i_k}$ in $M_m$.

*Case (i).* Suppose $V$ is not swapped with any other nodes. This means that there is a causal path $V_0' \rightarrow V_{i_{k_0}} \rightarrow V$ in $M_0$, while $V_0'$ is not connected to $V$. Besides, since the edge from $V_0$ to $V$ is added by replacing $V_{i_{k_0}}$ with $V_{j_{k_0}}$ in $M_{k_0}$, $V_0'$ is a parent of $V_{j_{k_0}}$ in $M_0$. If all added edges are cancelled out, then $V$ has at most $|Pa_{M_0}(V)|$ parents in $M_m$, i.e., $\text{TE}_{M_0}(An_{M_0}(V), V)$ can be written as the linear combination of at most $|Pa_{M_0}(V)|$ vectors in $\{\text{TE}_{M_0}(An_{M_0}(V), V') : V' \in Pa_{M_m}(V)\}$. Further, if no edges in $M_0$ are canceled out in $M_m$, then $V_{j_{k_0}}$ is a parent of $V$ in $M_m$. However, $V_{j_{k_0}}$ is a $\texttt{u-leaf}$ node in $M_0$ with parent set not being a subset of $Pa_{M_0}(V)$ (because $V_0' \in Pa_{M_0}(V_{j_{k_0}})$ and $V_0' \notin Pa_{M_0}(V)$). Therefore $M_0$ violates Assumption 2*(b1)*.

*Case (ii).* Suppose $V = V_{i_k}$ for some $k \in [m]$. This means that there is a causal path $V_0' \rightarrow V_{i_{k_0}} \rightarrow V_{i_k} \rightarrow V_{j_k}$ and a connection from $V_0'$ to $V_{j_{k_0}}$ in $M_0$. Further, there is a connection from $V_0$ to $V_{j_k}$ in $M_k$ which may or may not be an added edge. If all added edges are cancelled out, then $V_{j_k}$ has at most $|Pa_{M_0}(V_{i_k})|$ parents in $M_m$, i.e., $\text{TE}_{M_0}(An_{M_0}(V_{j_k}) \setminus \{V_{i_k}\}, V_{j_k})$ can be written as the linear combination of at most $|Pa_{M_0}(V_{i_k})|$ vectors in $\{\text{TE}_{M_0}(An_{M_0}(V_{j_k}) \setminus \{V_{i_k}\}, V') : V' \in Pa_{M_m}(V_{j_k})\}$. Note that $|Pa_{M_0}(V_{i_k})| \leq |Pa_{M_0}(V_{i_k}) \cup Pa_{M_0}(V_{j_k}) \setminus \{V_{i_k}\}|$, and the equality holds only if $Pa_{M_0}(V_{j_k}) \setminus \{V_{i_k}\} \subseteq Pa_{M_0}(V_{i_k})$, i.e., parents of $V_{j_k}$ in $M_0$ are all parents of $V_{i_k}$, except for $V_{i_k}$ itself. In this case, $V_{j_{k_0}}$ is a parent of $V_{j_k}$ in $M_m$, but it is a $\texttt{u-leaf}$ node in $M_0$ with the parent set not being a subset of $V_{i_k}$ (because of $V_0'$). Therefore $M_0$ violates Assumption 2*(b2)*.

*Case (iii).* Suppose $V = V_{j_k}$ for some $k \in [m]$. Note that the proof of Case (iii) in Part (2) of the necessity proof also applies here: If $V_0$ is not a parent of $V_{i_k}$ in $M_m$, then $V_0$ is not a parent of $V_{i_k}$ in $M_k$. According to Equations (13) and (19), this means that $V_0$ is not a parent of $V_{j_k}$ in $M_{k-1}$, which reduces to Case (i) and implies that $M_0$ violates Assumption 2*(b1)*.

In conclusion, if the ground-truth satisfies Assumption 2, then $M_m$ has at least one more added edges than $M_0$.

**Proof of (3)**: From Parts (1) and (2), we conclude that no edges are removed from $M_0$ in $M_m$, and there is at least one edge added in $M_m$ compared to to $M_0$. Combining both parts completes the proof of Proposition 4.

In the following we show that because of this additional edge, $M_m$ violates Assumption 2. Suppose the edge from $V_0$ to $V$ is added in $M_m$. Similar as Part (2), the position of $V_0$ might be changed. Denote $V_0'$ as the center of the ancestral ordered group of $V_0$ in $M_0$ (which may be $V_0$ itself). We consider each of the three following cases:

(i) $V$ is not swapped with any other nodes, i.e., $V \notin \{V_{i_1}, \cdots, V_{i_m}, V_{j_1} \cdots, V_{j_m}\}$. This means that $V_0'$ is not a parent of $V$ in $M_{k_0}$, but $V_0$ is a parent of $V$ in $M_m$.

(ii) $V = V_{i_k}$ for some $k \in [m]$. This means that $V_0'$ is not a parent of $V_{i_k}$ in $M_{k_0}$, but $V_0$ is a parent of $V_{j_k}$ in $M_m$.

(iii) $V = V_{j_k}$ for some $k \in [m]$. This means that $V_0'$ is not a parent of $V_{j_k}$ in $M_{k_0}$, but $V_0$ is a parent of $V_{i_k}$ in $M_m$.

*Case (i).* Suppose $V$ does not change its position. If the edge from $V_0'$ to $V$ is added in $M_m$, then $V$ has at least one more parent in $M_m$ than in $M_0$, i.e., $|Pa_{M_m}(V)| > |Pa_{M_0}(V)|$. Recall that in the ground-truth, $\text{TE}_{M_0}(An_{M_0}(V), V)$ can be written as the linear combination of $|Pa_{M_0}(V)|$ vectors in $\{\text{TE}_{M_0}(An_{M_0}(V), V') : V' \in Pa_{M_0}(V)\}$, where parents of $V$ in $M_0$ are all possible parents of $V$ in $M_m$. Further, for all $V' \in Pa_{M_0}(V) \cup \{V\}$, $\text{TE}_{M_0}(An_{M_0}(V), V') = \text{TE}_{M_m}(An_{M_m}(V), V')$. Therefore, $\text{TE}_{M_m}(An_{M_m}(V), V)$ can be written as the linear combination of strictly less than $|Pa_{M_m}(V)|$ vectors in $\{\text{TE}_{M_m}(An_{M_m}(V), V') : V'$ is a possible parent of $(V)$ in $M_m\}$, which means that $M_m$ violates Assumption 2*(b1)*.

*Case (ii).* Suppose $V = V_{i_k}$ for some $k \in [m]$. In this case, $V_{j_k}$ has at least one more parents in $M_m$ than $V_{i_k}$ has in $M_0$, i.e., $|Pa_{M_m}(V_{j_k})| > |Pa_{M_0}(V_{i_k})|$. Further, for all $V' \in Pa_{M_0}(V_{i_k}) \cup \{V_{i_k}\}$, $\text{TE}_{M_m}(An_{M_m}(V_{i_k}) \setminus \{V_{j_k}\}, V') = \text{TE}_{M_0}(An_{M_0}(V_{i_k}), V')$, and parents of $V_{i_k}$ in $M_0$ are all possible parents of $V_{j_k}$ in $M_m$. Therefore, $\text{TE}_{M_m}(An_{M_m}(V_{i_k}) \setminus \{V_{j_k}\}, V_{i_k})$ can be written as the linear combination of $|Pa_{M_0}(V_{i_k})| < |Pa_{M_m}(V_{j_k}) \cup Pa_{M_m}(V_{i_k}) \setminus \{V_{j_k}\}|$ vectors in $\{\text{TE}_{M_m}(An_{M_m}(V), V') : V'$ is a possible parent of $V_{j_k}$ in $M_m\}$, which means that $M_m$ violates Assumption 2*(b2)*.

*Case (iii).* Suppose $V = V_{j_k}$ for some $k \in [m]$. In this case, $|Pa_{M_m}(V_{i_k})| > |Pa_{M_0}(V_{j_k})|$. Recall that $\text{TE}_{M_0}(An_{M_0}(V_{j_k}), V_{j_k})$ can be written as the linear combination of $|Pa_{M_0}(V_{j_k})|$ vectors in $\{\text{TE}_{M_0}(An_{M_0}(V_{j_k}), V') : V' \in Pa_{M_0}(V_{j_k})\}$. Since $V_{j_k}$ is a $\texttt{u-leaf}$ node in $M_0$, $\text{TE}_{M_0}(An_{M_0}(V_{j_k}), V_{i_k})$ can be written as the linear combination of $|Pa_{M_0}(V_{j_k})|$ vectors in $\{\text{TE}_{M_0}(An_{M_0}(V_{j_k}), V') : V' \in Pa_{M_0}(V_{j_k}) \cup \{V_{j_k}\} \setminus \{V_{i_k}\}\}$. Note that for all $V' \in Pa_{M_0}(V_{j_k}) \cup \{V_{j_k}\}$, $\text{TE}_{M_0}(An_{M_0}(V_{j_k}), V') = \text{TE}_{M_m}(An_{M_m}(V_{i_k}), V')$, and parents of $V_{j_k}$ in $M_0$ other than $V_{i_k}$ are all possible parents of $V_{i_k}$ in $M_m$. Therefore, $\text{TE}_{M_m}(An_{M_m}(V_{i_k}), V_{i_k})$ can be written as the linear combination of

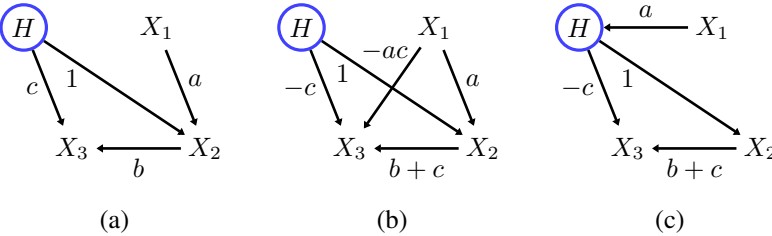

(a)            (b)            (c)

Figure 6: Comparison between our identifiability condition and the results in [1]. (a) The ground-truth of a linear SEM-UR with every edge satisfying Condition 2, but the model violates the conditions in [1]. (b) The alternative linear SEM-UR in the AOG equivalence class, which violates Assumption 3*(b)*. (c) An alternative model (considered in [1]) with the latent variable $H$ being non-root.

$|Pa_{M_0}(V_{j_k}) \cup \{V_{j_k}\} \setminus \{V_{i_k}\}| = |Pa_{M_0}(V_{j_k})| < |Pa_{M_m}(V_{i_k})|$ vectors in $\{\text{TE}_{M_m}(An_{M_m}(V_{i_k}), V') : V'$ is a possible parent of $(V_{i_k})$ in $M_m\}$, which means that $M_m$ violates Assumption 2*(b1)*.

To summarize, if any edge is added to the model, then this edge must cause certain violation of Assumption 2.

**Conclusion**: We showed that by replacing some `u-leaf` nodes in the ground-truth $M_0$ by some other nodes in ancestral ordered group of each `u-leaf` node, where at least one of them is not in the same different direct ordered group of the `nu-leaf` node, the alternative model $M_m$ (1) has the same mixing matrix, (2) has at least one more edge than the graph structure of $M_0$, and (3) violates Assumptions 2. Therefore we can distinguish $M_0$ from any other model outside the DOG equivalence class.

Combining the necessity part and the sufficiency part together, we conclude that DOG equivalence class is the extent level of identifiability under Assumption 1 and 2.

## D    Comparison with partially observed linear causal models

The linear SEM-UR defined in Definition 3 is a subclass of the linear SEM with latent variables proposed in [1]. Specifically, [1] does not require the latent variables to be parent-less. Further, the linear SEM-ME in Definition 2 can also be viewed as a subclass of the model in [1], where the latent variables correspond to unobserved underlying variables, $\mathcal{Z}$, and the observed variables correspond to observed underlying variables $\mathcal{Y}$ and the measurements of $\mathcal{Z}$ (i.e., $\mathcal{U}$).

The authors of [1] derived necessary conditions for unique identifiability of the model with the fewest number of edges, which are also sufficient together with "bottleneck faithfulness assumption" (cf. Condition 3). The derived conditions are two-fold: (1) The bottleneck condition, which requires the children set of any (observed or latent) variable $V_i$ to be the smallest set to block all directed paths from $V_i$ to its observed descendants; and (2) The strong non-redundancy condition, which requires that the children set of any latent variable $H_i$ not to be a subset of that of any other variable $V_i$, plus $V_i$ itself.

Recall that from Proposition 4, our identifiability condition also guarantees that if the ground-truth is uniquely identifiable, then any other model that has the same mixing matrix as the ground truth and satisfies conventional faithfulness has strictly more edges. We have shown in Appendix B.4 that our SEM-UR assumption is implied from bottleneck faithfulness assumption. Further, our condition for unique identifiability of a SEM-UR is also implied from from the identifiability conditions in [1]. Specifically, for the linear SEM-UR, the bottleneck condition is automatically satisfied, and the strong non-redundancy condition implies Condition 2(a). In contrast, our unique identifiability condition for the linear SEM-UR does not imply the conditions in [1]: A linear SEM-UR satisfying Condition 2 may not be uniquely identifiable in [1]. This is because the model considered in [1] is a larger class than the SEM-UR. An example is shown in Figure 6, where the linear SEM-UR generated according to Figure 6(a) can be uniquely identified (as a SEM-UR). The only other model in its AOG equivalence class is shown in Figure 6(b) (by changing the exogenous noise term associated with $X_2$ from $N_{X_2}$ to $N_H$), which violates Assumption 3 and also has more edges than (a). However, the ground-truth is not uniquely identifiable in [1], where an alternative model is shown in Figure 6(c). Note that Figure 6(c) and Figure 6(a) have the same number of edges, and hence [1] cannot distinguish it from the ground-truth. On the contrary, Figure 6(c) is not a SEM-UR since $H$ has a parent, and can distinguished from the ground-truth. Note that Figure 6(a) and 6(c) have exactly the same total causal effects among all pairs of observed variables.

Finally, we note that if we were to apply the identifiability result in [1] to the SEM-ME, this leads to a trivial special case. Specifically, if the nun-redundancy condition is satisfied, then all leaf nodes must be observed (i.e., there is no `u-leaf` node in the model). The reasoning behind this is as follows. If there is an unobserved leaf node $Z_i$ with its measurement $U_i$, we can remove $Z_i$, and draw edges from the parents of $Z_i$ to $U_i$. The resulting model has the same mixing matrix and one less edge ($Z_i \rightarrow U_i$ removed). However, this corresponds

to a trivial special case of SEM-ME: If there is no `u-leaf` node, then the SEM-ME can always be uniquely identified, since each `nu-leaf` node is in a separate ancestral ordered group. Consequently, our identifiability results for the linear SEM-ME do not follow from the identifiability results in [1].

# E Numerical experiments

We evaluated the performance of our recovery algorithm on randomly generated linear SEM-MEs and SEM-URs with different number of variables. We considered two cases: (1) when a noisy version of the mixing matrix is given, where the noise is Gaussian with different choices of variance denoted by $\mathtt{d}^2$, and (2) when synthetic data comes from a linear generating model with non-Gaussian noises with different sample sizes.

## E.1 Simulation settings

**Model generation.** We randomly generate synthetic linear SEM-MEs and SEM-URs as follows. For SEM-ME, we first generate a linear SEM with $p$ underlying variables, where each pair of observed variables is connected with probability $p_e$. Then, we randomly select some of the $p$ underlying variables to be unobserved (measured with error), where the measurement error is added after the coefficients are drawn; note that we only add measurement error to the unobserved variables corresponding to the non-leaf nodes. The corresponding mixing matrix is calculated (i.e., $(\mathbf{I} - \mathbf{A})^{-1}$), where $\mathbf{A}$ is the adjacency matrix among underlying variables in $\mathcal{Z} \cup \mathcal{Y}$. For SEM-UR, we generate a SEM with $q$ observed variables and $l$ latent variables. Each pair of observed variables, and each pair of latent and observed variables, is connected with probability $p_e$. We ensure that each latent variable has at least two children. The strength (coefficients) of the edges in both models are independently drawn from the uniform distribution $[0.5, 1]$. After the mixing matrix is obtained, we randomly permute its rows and columns to hide the true causal order.

**Baseline methods.** We compare the performance of our recovery algorithm (which we denote by DOG) with the following methods: (1) AOG-based method (AOG): After recovering the AOG from the mixing matrix, we randomly choose the centers in each ancestral ordered group. We then recover the full model based on the selection of the centers. This approach is similar to the recovery method in [19]. (2) ICA-LiNGAM [21]. [7] Note that ICA-LiNGAM method is not designed for systems with latent variables, but benefits from strong performance for recovering the mixing matrix in causally sufficient settings. The goal of this comparison is to demonstrate the necessity of using methods designed specifically to handle latent variables.

We compare with AOG both in Case (1) and Case (2), and we compare with ICA-LiNGAM only in Case (2).

**Metrics.** We compare the recovered matrices with the ground-truth matrix based on the following metrics:

(1) SHD / Edge: The normalized structural Hamming distance divided by the number of true edges in the graph;

(2) Precision: The fraction of edges in the recovered model that is also in the ground-truth;

(3) Recall: The fraction of edges in the ground-truth that can be recovered;

(4) F1 Score: The harmonic mean of Precision and Recall.

In linear SEM-ME, since we can only recover the ground-truth model up to its DOG equivalence class, we also compared the recovered matrices with the matrix that has the smallest $l_2$ distance (Frobenius norm) among all models in the DOG equivalence class of the ground-truth. Furthermore, In linear SEM-UR, DOG and AOG methods return both the adjacency matrix $\mathbf{A}$ among observed variables and the adjacency matrix $\mathbf{B}$ from latent to observed variables. On the contrary, ICA-LiNGAM only returns the matrix $\mathbf{A}$. Therefore, we compare the performance of all three methods in recovering $\mathbf{A}$, and we compare the performance of DOG and AOG methods in recovering $\mathbf{B}$. Note that $\mathbf{B}$ can only be learned up to permutation and scaling of the columns.

**Pre-processing of the mixing matrix.** In linear SEM-ME, the mixing matrix used in recovery ($\mathbf{W}^{ME}$) is a submatrix of the overall mixing matrix $\mathbf{W}$. Given $\mathbf{W}$ and the indices corresponding to the unobserved variables, we extract $\mathbf{W}^{ME}$ from $\mathbf{W}$ as follows. We first normalize each column by the entry with the largest absolute value. Next, for each index of an unobserved variable $i$, we find the column vector in $\mathbf{W}$ that has the smallest $l_2$-distance to the one-hot vector $e_i$ (i.e., only the $i$-th entry is 1), and remove that column from $\mathbf{W}$. We repeat until all one-hot vectors of the measurement errors are removed.

We note that the recovery of AOG from the mixing matrix $\mathbf{W}^{ME}/\mathbf{W}^{UR}$ (cf. Algorithm 1) requires that we can always find a row/column with only one non-zero entry in each iteration of the algorithm. To ensure that the recovered mixing matrix satisfies this property, we sort all non-zero entries in the mixing matrix with respect to their absolute values (from smaller to larger), and iteratively remove the entries with the smallest absolute values until a valid AOG is returned.

---

[7] `https://sites.google.com/view/sshimizu06/lingam`

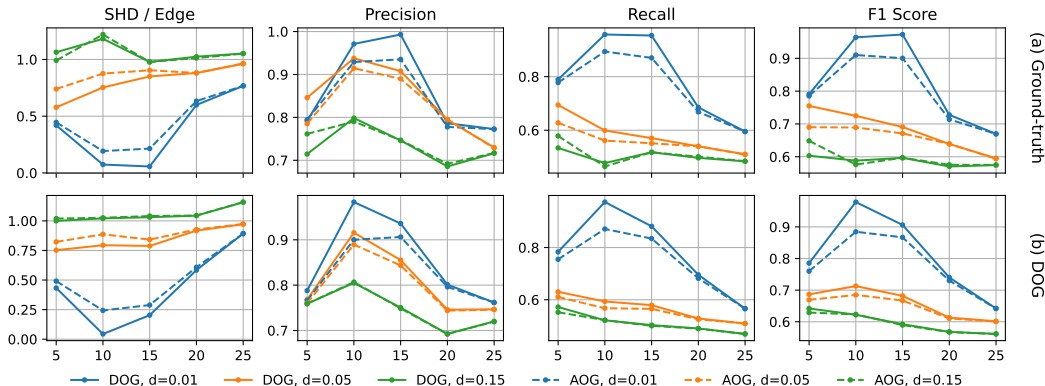

Figure 7: Performance of SEM-ME recovery given the noisy mixing matrix. $x$-axis represents the number of total variables ($p$). The first row compares the recovered matrix with the ground-truth, and the second row compares the recovered matrix with its closest in the DOG equivalence class of the ground-truth.

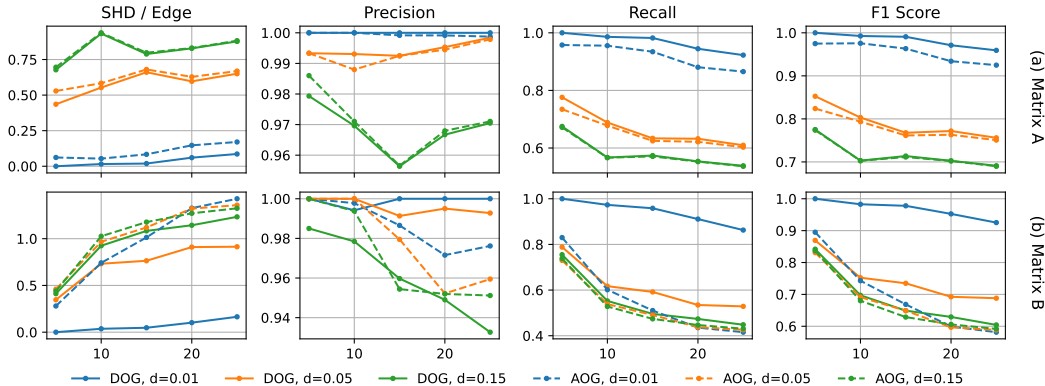

Figure 8: Performance of SEM-UR recovery given the noisy mixing matrix. $x$-axis represents the number of total variables ($q + l$). The first row compares the recovery of matrix $\mathbf{A}$, and the second row compares the recovery of matrix $\mathbf{B}$.

## E.2  Noisy mixing matrix

We test the performance of our method for Case (1), i.e., when a noisy version of the mixing matrix is given, . Given the true mixing matrix, the noisy matrix is constructed by first adding an independent Gaussian noise with variance $\mathtt{d}^2$ to each non-zero entry of $\mathbf{W}$, and then adding independent Gaussian noise with variance $\mathtt{d}^2$ to each entry of $\mathbf{W}$ with probability $0.2$. $\mathtt{d}$ is the parameter controlling the noise variance, which is selected as $[0.01, 0.05, 0.15]$.

For linear SEM-ME, We select $p = [5, 10, 15, 20, 25]$, and we randomly select $80\%$ of the variables to be measured with error. We set $p_e = 0.4$, and repeat the simulation for $50$ times. The average performance is shown in Figure 7. For linear SEM-UR, We select $q = [4, 8, 12, 16, 20]$, and we select $l = [1, 2, 3, 4, 5]$, respectively. We set $p_e = 0.4$, and repeat the simulation for $50$ times. The average performance is shown in Figure 8.

We observe that our DOG method performs better than AOG method for the recovery of both models. Besides, the performance of both algorithms are better if the variance $\mathtt{d}$ is small. Further, Recall from Corollary 3 that all SEM-URs in the same DOG have the same structure. This can be seen in Figure 8, where with $\mathtt{d} = 0.01$, our DOG algorithm has a high accuracy in recovering the graph structure of the ground-truth. As for the recovery of SEM-ME in Figure 7, we observe that even though theoretically the structure of the ground-truth model can only be identified up to the DOG equivalence class, our DOG algorithm can recover part of the ground-truth structure that is shared among the DOG equivalence class.

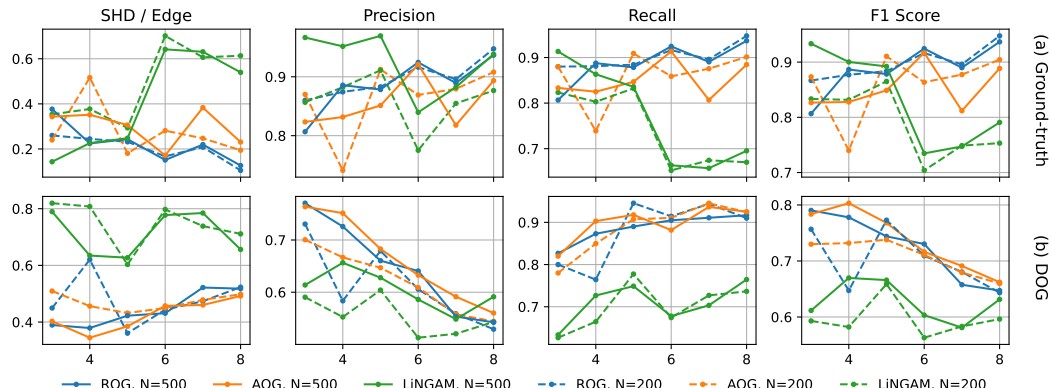

Figure 9: Performance of SEM-ME recovery based on synthetic data. $x$-axis represents the number of observed variables ($p$). The first row compares the recovered matrix with the ground-truth, and the second row compares the recovered matrix with its closest in the DOG equivalence class of the ground-truth.

### E.3 Synthetic data with non-Gaussian noises

We test the performance of our recovery algorithm for Case (2), where the synthetic data is generated from the linear causal model with non-Gaussian exogenous noise terms, and the sample size is proportional to the number of observed variables. That is, the sample size can be expressed as $\mathbb{N}p$ (resp. $\mathbb{N}q$), where $p$, $q$ are the number of observed variables in both models, and $\mathbb{N}$ is selected to be $\{200, 500\}$. In this case, the mixing matrix needs to be estimated using ICA methods. Note that the overall mixing matrices in both problems have more columns than rows, therefore we need to use an overcomplete ICA method. The overcomplete ICA method that we use is Reconstruction ICA [14]. We assume that the number of sources is known for overcomplete ICA.

In linear SEM-ME, we select $p = 3, 4, 5, 6, 7, 8$. We assume that when $p = 3, 4, 5$, there is only one unobserved non-leaf node, and each leaf node is observed with probability 0.5. Under all three settings, the overall mixing matrix is of dimension $p \times (p + 1)$. Similarly, when $p = 6, 7, 8$, there are two unobserved non-leaf nodes, and each leaf node is observed with probability 0.5. In linear SEM-UR, we select $q = 3, 4, 5, 6, 7, 8$. Similar to SEM-ME, we assume that when $q = 3, 4, 5$, there is only one latent confounder, and the overall mixing matrix is of dimension $q \times (q + 1)$. When $p = 6, 7, 8$, there are two latent confounders in the system. We set $p_e = \min(0.5, 2.5/(p - 1))$ (resp. $2.5/(q - 1)$) in both models, and the exogenous noises are independently drawn from uniform distributions between $[-0.5, 0.5]$.

We use bootstrapping method [6] to improve the stability of the recovery of overcomplete ICA with 50 iterations. In each iteration, we randomly select $80\%$ of the sampled data, and get an estimate of the mixing matrix using Reconstruction ICA. After each iteration, we repermute and rescale the columns of the recovered mixing matrix such that they are aligned with the mixing matrix recovered in the first iteration. Lastly, we use hypothesis testing to prune the recovered mixing matrix and remove the entries whose absolute value is below a threshold of 0.1 in SEM-UR and 0.2 in SEM-ME, with $95\%$ confidence level.

We compared our DOG algorithm with AOG and ICA-LiNGAM for both models (SEM-ME and SEM-UR), and we record the average of the performance metrics after 50 graphs are generated for each model. The performance of recoveries for linear SEM-ME and SEM-UR are shown in Figure 9 and Figure 10, respectively. We observe that the performance of ICA-LiNGAM depends on the number of unobserved variables/latent confounders: There is a sharp decrease in the performance of ICA-LiNGAM when we increase the number of unobserved variables/latent confounders (from 1 unobserved/latent variable for $p = 3, 4, 5$ to 2 unobserved/latent variables for $p = 6, 7, 8$). On the contrary, DOG and AOG methods outperform ICA-LiNGAM for both models. This implies that developing methods for latent variable models is necessary. Besides, our DOG method outperforms AOG methods for both models, which demonstrates the effectiveness of our proposed algorithm.

### E.4 Plots of the error bars

Figure 4 in the main text includes three algorithms and multiple experimental settings. We chose not to include error bars in that figure so that the presentation does not become cluttered. Instead, here, we report the error bars for the simulation results in Figure 4, for all three algorithms.

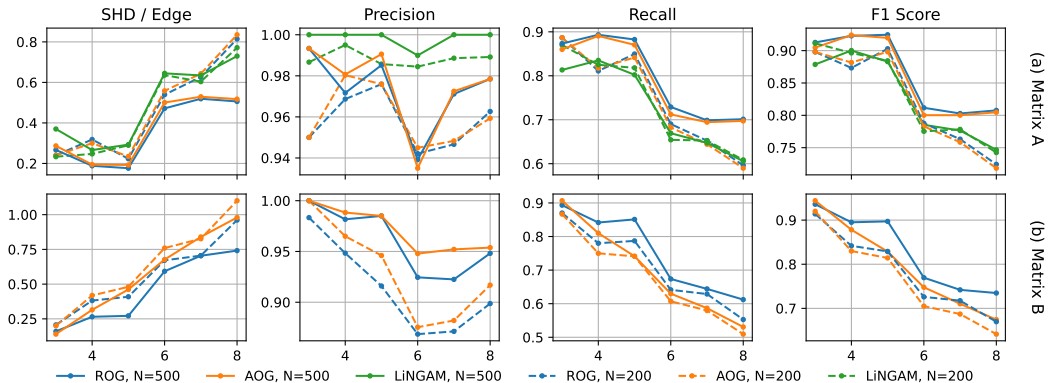

Figure 10: Performance of SEM-UR recovery based on synthetic data. $x$-axis represents the number of observed variables ($q$). The first row compares the recovery of matrix $\mathbf{A}$, and the second row compares the recovery of matrix $\mathbf{B}$. Recall that ICA-LiNGAM only returns matrix $\mathbf{A}$ as it assumes causal sufficiency.

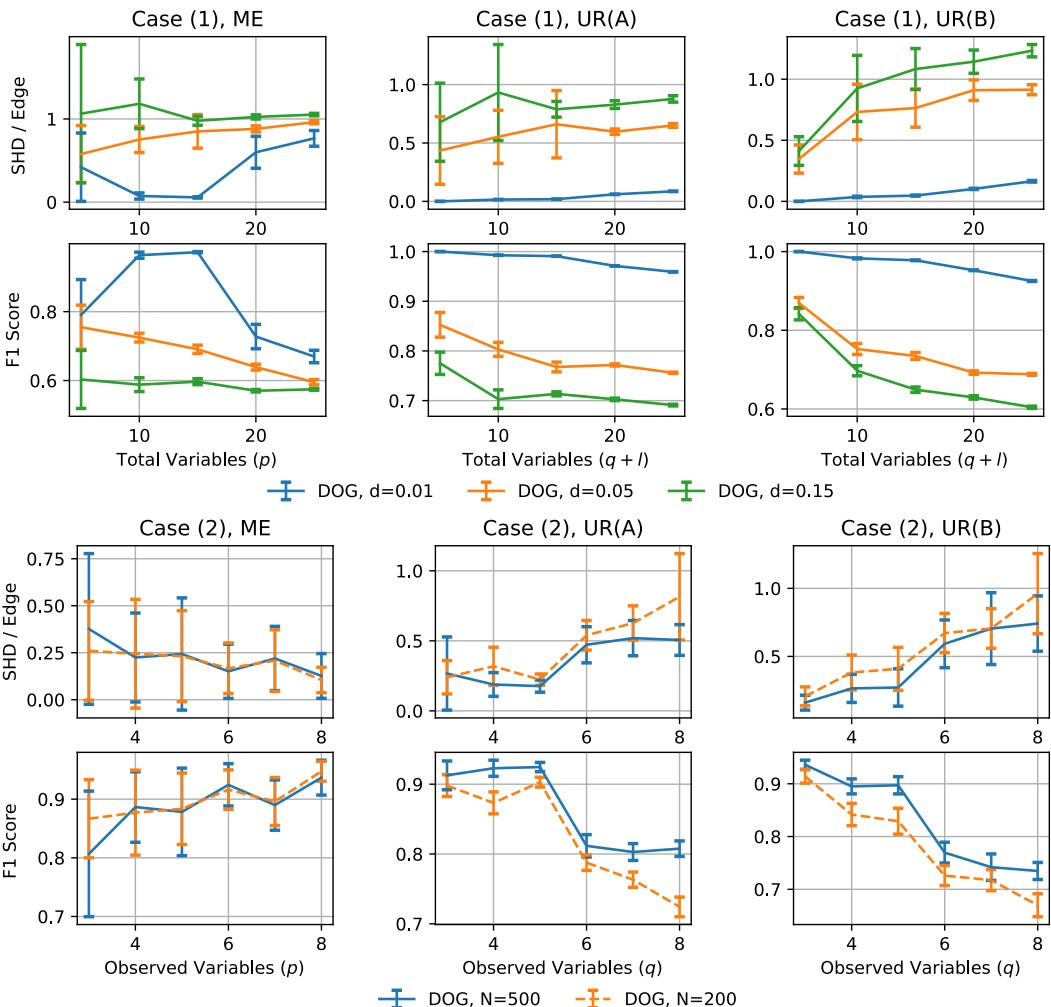

Figure 11: Error bars of DOG algorithm under different settings of the experiment, namely, Case (1) in which we have a noisy mixing matrix; Case (2) where we use raw observational data; SEM-ME recovery; recovery of matrices $\mathbf{A}$ and $\mathbf{B}$ for SEM-UR, and different metrics, SHD/Edge and F1 score.

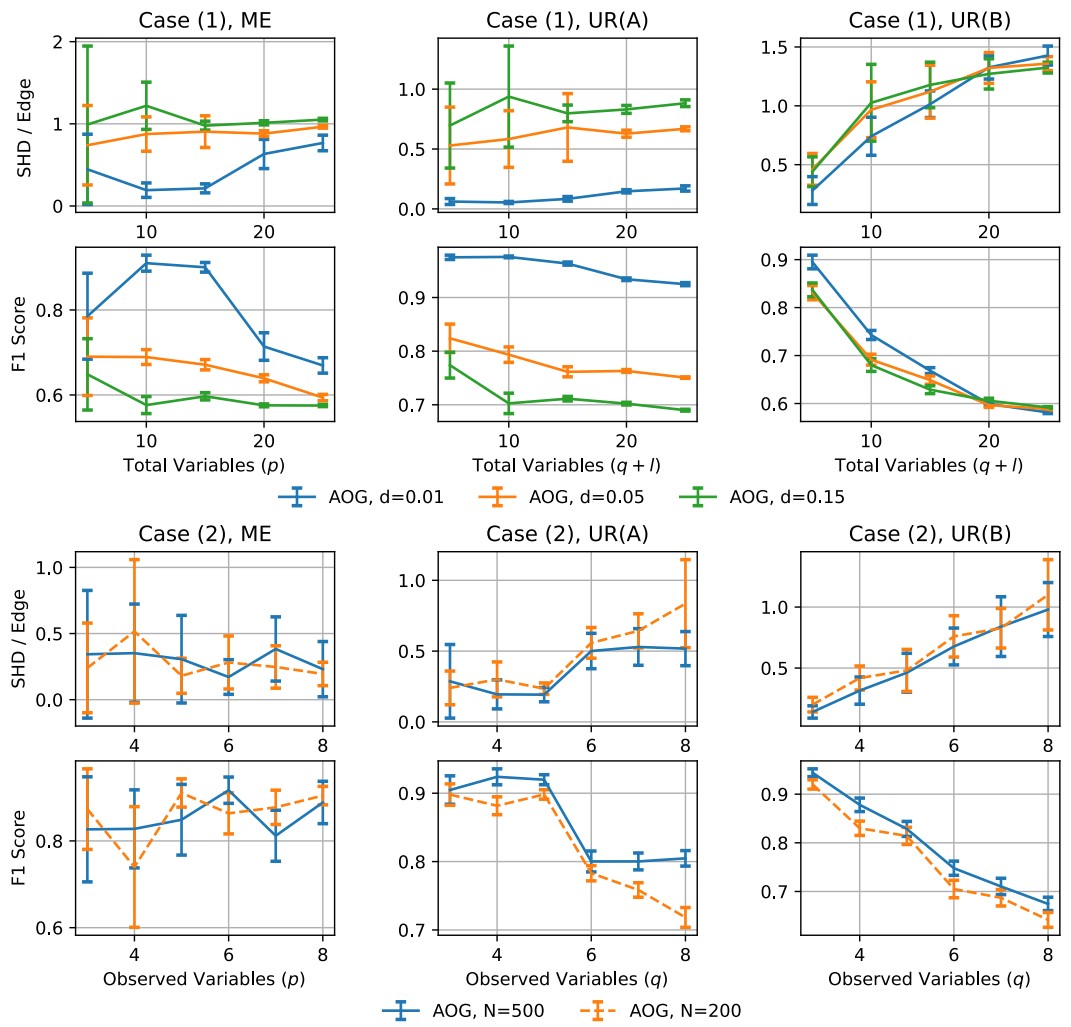

Figure 12: Error bars of the AOG algorithm under different settings of the experiment.

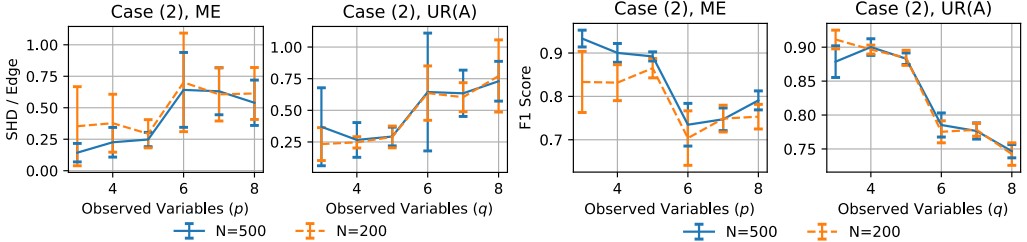

Figure 13: Error bars of ICA-LiNGAM algorithm under different settings of the experiment. Note that since ICA-LiNGAM assumes causal sufficiency, the mixing matrix in ICA-LiNGAM must be a square matrix, i.e., it does not recover matrix $\mathbf{B}$ for SEM-UR. Additionally, Case (1), i.e., noisy mixing matrix, cannot be applied here, since the true mixing matrix is not square.