# OpenReview forum: "Causal Discovery in Linear Latent Variable Models Subject to Measurement Error"
_NeurIPS.cc/2022/Conference — NeurIPS 2022 Accept_

### Official Review · Reviewer_KoAc · 2022-06-28

**Rating:** 7
**Confidence:** 3
**Soundness:** 3 good
**Presentation:** 4 excellent
**Contribution:** 3 good

**Summary:**

The article studies causal discovery in linear systems where some of the variables are observed with measurement error. The identification is based on the mixing matrix (which maps independent exogenous noise to the observations). The article first shows that the model with measurement error is equivalent to a linear latent variable model where the latent variables don't have any parents, so any identifiability results based on the mixing matrix are applicable to both cases (Thm 1).

Thm 2 states that under the regular faithfulness assumption the models can be identified up to an equivalence class consisting of systems with the same ancestral ordered grouping (AOG), but the edges between the groups can't be identified. Then the article presents an additional condition (Condition 1) such that iff every edge from a non-leaf node satisfies the condition, then even the edges between the groups can be identified, in other words, the system can be identified up to the Directed Ordered Grouping (DOG).

Furthermore, the assumptions imply that any ground-truth generating model has fewer edges than any model that has the same mixing matrix but does not belong to the same DOG as the true model. This insight can be used to devise an algorithm for structure recovery: 1) find mixing matrix, 2) find AOG of the true model, 3) find the center for each group that leads to the graph with fewest edges.

A simple simulation study shows that using the algorithm with DOG identifies the correct model better than recovery based on just AOG or another published baseline: ICA-LiNGAM.


**Questions:**

Figure 4 could benefit from additional annotation of the rows/columns and a more detailed caption.

**Limitations:**

Yes.

**Strengths And Weaknesses:**

Strenghts:

The presentation is clear for this kind of work that is somewhat heavy on technical detail. All the math is presented accurately (though I only glanced through the supplement) and the theorems are associated with illustrative figures and examples that demonstrate the concepts in concrete setups.

The main novelty are the theoretical results connecting the two types of graphs as well as the new results on identifiability that are stronger than the previous ones. I find these results original and significant in the sub-field machine learning studying causal identifiability with latent variable models.

Weaknesses:

There are no major weaknesses. Some real-world example would have strengthened the paper further, but that might not be necessary in this kind of paper where the main contribution are the new theoretical results.

One major assumption for identifiability is that the noise terms are non-Gaussian. Whereas this might be true in many real-world situations, it is equally true that Gaussian noise is the default assumption in the majority of applications. Hence, the assumption may be limiting.

---

> ### Author Response · Authors · 2022-08-02
> **Response to Reviewer KoAc**
>
> We thank the reviewer for the comments. We address the reviewer's concerns below.
>
> **Real world examples of the considered model:**
>
> We will add the following discussion of the applications of our work to the revised paper:
>
> In most real-life applications, we do not observe the exact value of some of the variables and we can only have measurements of these underlying variables that are prone to errors. For example, in neuroscience and genomics, the measured brain signals obtained by functional magnetic resonance imaging (fMRI) and the measured gene expression using RNA sequencing usually contain error through the measurement process [16, 22]. Another example is the responses to psychometric questionnaires where the questions represent noisy views of various traits [1; Scheines and Ramsey, 2016]. Moreover, in most real-life setups we are not able to observe or measure many variables in the system (for instance in socioeconomical applications many variables are unobserved or even unknown) and hence, the violation of the causal sufficiency assumption is the norm rather than the exception.
> For example, in the field of non-invasive neuroimaging such as Electroencephalography (EEG), the activities of many of the brain regions are not recorded. Hence causal sufficiency cannot be assumed [Mastakouri et al, 2019]. Similarly, in stock market, returns may be confounded by a several unmeasured economic and political factors [1].
>
> **Regarding the assumption of non-Gaussianity of the noise terms:**
>
> We agree with the reviewer that Gaussian noise is the default assumption in most works. Yet, as the reviewer said, the assumption of non-Gaussian noises in fact holds in many real-world situations such as brain EEG signals and stock returns described above [17; Hyvärinen et al, 2010].
>
> However, we would like to emphasize that the results presented in our work are not just limited to non-Gaussian noise. Indeed, our results can be applied to any model satisfying the separability assumption (Assumption 1), and non-Gaussanity is a special case where (with an extra mild assumption for the UR model and no extra assumptions for the ME model) this assumption is satisfied. For example, besides the non-Gaussian noise setting, the mixing matrix can be recovered when (1) the observed data are sampled from several domains (or time index), and the distribution of the exogenous noise changes across domains/over time, while satisfying certain mild condition [1]; or (2) the exogenous noises are piecewise constant functions satisfying another set of mild conditions [2].
>
> Nevertheless, non-Gaussian generating model is one of the main settings for which separability assumption holds. However, this model is a less-studied part of the literature and the extent of identifiability for Gaussian models is mostly known. Specifically, a linear Gaussian SEM can only be recovered up to its Markov Equivalence class (with or without latent confounding), and it can only be uniquely identified under further assumptions, e.g., sparsity [Raskutti and Uhler, 2018] or equal variance [Peters and Bühlmann, 2014]. On the contrary, assuming non-Gaussian noise, the model can be uniquely recovered under causal sufficiency. Yet, when latent confounders exist, the exact extent of identifiability was previously unknown. We showed that in this setting, under our assumptions, the extent of identifiability is characterized by AOG/DOG equivalence classes.
>
> **Regarding Figure 4:**
>
> In the revised paper, we will:
>
> 1. Move the experiment details and explanation of the results from Appendix D to the main text.
> 2. Add more details to the caption of Figure 4. For instance, lower SHD/Edge value and higher F1-score indicate better performances.
>
>
>
> We will upload revised supplementary material which includes the modifications listed above. These modifications appear in Appendix E, and will be implemented in the camera-ready draft if accepted.
>
> **References:**
>
> Scheines R, Ramsey J. Measurement error and causal discovery[C]//CEUR workshop proceedings. NIH Public Access, 2016, 1792: 1.
>
> Mastakouri A, Schölkopf B, Janzing D. Selecting causal brain features with a single conditional independence test per feature[J]. Advances in Neural Information Processing Systems, 2019, 32.
>
> Hyvärinen A, Zhang K, Shimizu S, et al. Estimation of a structural vector autoregression model using non-gaussianity[J]. Journal of Machine Learning Research, 2010, 11(5).
>
> Raskutti G, Uhler C. Learning directed acyclic graph models based on sparsest permutations[J]. Stat, 2018, 7(1): e183.
>
> Peters J, Bühlmann P. Identifiability of Gaussian structural equation models with equal error variances[J]. Biometrika, 2014, 101(1): 219-228.

---

> > ### Comment · Reviewer_KoAc · 2022-08-09
> > **Thanks**
> >
> > Thank you for your clarifications. I found the discussion on envisioned real-world applications and the non-Gaussianity assumption clear and useful. I also benefitted from the "impact of our work on causal discovery" that you wrote in response to another reviewer's comments. Expanding the paper's introduction or discussion along these lines as well as improving the clarity of the experimental results will be sufficient, in my opinion, that the paper can be accepted, and I will consequently improve my score to 7. That said, a good real-world demonstration would have increased the significance even further.

---

> > > ### Author Response · Authors · 2022-08-09
> > > **Response**
> > >
> > > We thank the reviewer for the response. We will definitely implement these changes if the paper is accepted.

---

> ### Comment · Area_Chair_RvW8 · 2022-08-08
> **Response needed**
>
> Dear Reviewer KoAc,
>
> Please kindly respond to the rebuttal provided by the authors and/or engage in the discussion with them. If it addresses your concerns, please react accordingly. Otherwise, please elaborate in your review on why you think the rebuttal/discussion is inadequate. Thank you.
>
> Best, AC

---

### Official Review · Reviewer_C5rx · 2022-07-07

**Rating:** 6
**Confidence:** 4
**Soundness:** 4 excellent
**Presentation:** 3 good
**Contribution:** 2 fair

**Summary:**

The paper shows a correspondence between linear SEMs with measurement error and linear SEMs with hidden variables.

The paper then characterizes the equivalence class for these cases which can be learned when variables are linear and non-Gaussian under two faithfulness assumptions.

A recovery algorithm is provided along with simulation results, which is shown to outperform LiNGAM.

**Questions:**

What is the primary impact of these results on causal discovery broadly?

What are some applications that these results contribute to?

**Limitations:**

Some discussion of limitations.

**Strengths And Weaknesses:**

Strengths
- Rigorous theoretical results that contribute to the literature on identifiability of linear non-Gaussian models
- Very detailed and generally well written paper

Weaknesses
- Very limited discussion of motivation and impact of the work
- Limited empirical results and extremely limited discussion of empirical results

---

> ### Author Response · Authors · 2022-08-02
> **Response to Reviewer C5rx (Part 1/2)**
>
> We thank the reviewer for the comments. We address the reviewer's concerns below.
>
> **Impact of our work on causal discovery:**
>
> The impact of this work on causal discovery can be summarized as follows:
>
> 1. We build a mapping between linear SEM-ME and SEM-UR in Theorem 1, and show that any identifiability results for one model (based on the mixing matrix) can be translated to the other. Specifically, we note that linear SEM-UR has been widely studied in the literature, while only a few works considered linear SEM-ME. Therefore, we believe that this mapping can be utilized to fill the gaps for the less studied model.
> 2. We show in Theorem 2 and Theorem 3 that under different faithfulness assumptions, the model can only be identified up to AOG and DOG equivalence class, respectively. That is, this result provides the extent of identifiability.  This can serve as a basis for characterizing theoretical guarantees such as consistency of new algorithms.
> 3. We prove that only separability and SEM-UR faithfulness conditions suffice for unique identifiability of the structure for linear SEM-UR models. This significantly relaxes the previous set of conditions in the literature [1, 17; Wang and Drton, 2020; Chen et al, 2021] that were shown to be sufficient for unique identifiability of the structure.
> 4. We show in Proposition 1 that under our faithfulness assumptions (Assumptions 2 and 3), the true generating model is always sparser than any other model that has the same mixing matrix but does not belong to the same DOG equivalence class.  This serves as a ground for positing sparsity assumption that appears frequently in the literature without rigorous justification.
>
> In the revised paper, we will add a discussion regarding the aforementioned impacts of our work on causal discovery literature.
>
> **Regarding empirical results:**
>
> Section 5 only includes part of the simulations, and full simulation results and discussions appear in Appendix D, page 22-25. Specifically, we compared the recovery of linear SEM-ME and SEM-UR under two cases (noisy mixing matrix and raw data), three algorithms (DOG, AOG, ICA-LiNGAM), and four metrics (SHD/Edge, Precision, Recall, F1 Score).
>
> Further, recall that our theoretical results show that the structure of a linear SEM-UR can be uniquely recovered, and the structure of a linear SEM-ME can be identified up to its DOG equivalence class. Therefore, for linear SEM-ME, we compared the recovered model with: (1) the ground-truth, and (2) its closest model in the DOG equivalence class of the ground-truth (in terms of Frobenius distance). For linear SEM-UR, we compared the recovery of the adjacency matrices (1) among observed variables (i.e., $\mathbf{A}$) and (2) from latent variables to observed variables (i.e., $\mathbf{B}$). The results show that our DOG-based method outperforms the baseline methods, and the performance is consistent with the theoretical analysis.
>
> We believe we have included more empirical results compared to the existing work which we have extended in terms of the theoretical results [1, 17, 22]. Please refer to Appendix D for more details about the simulation results. We will move some discussions from Appendix D to the main text in the revised paper.

---

> > ### Comment · Reviewer_C5rx · 2022-08-05
> > **Author response**
> >
> > I appreciate the authors' lengthy responses. While this partially addresses some concerns, I still have concerns about the empirical results and motivations as presented in the paper itself. I have increased my score to weak accent. If the paper is accepted I would encourage the authors to bring more of the empirical results into the text of the paper itself and not really so much on the supplement for evaluation of the method.

---

> > > ### Author Response · Authors · 2022-08-06
> > > **Response**
> > >
> > > We thank the reviewer for the response. We will definitely do so if the paper is accepted.

---

> ### Author Response · Authors · 2022-08-02
> **Response to Reviewer C5rx (Part 2/2)**
>
> **Application of the results:**
>
> In most real-life applications, we do not observe the exact value of some of the variables and we can only have measurements of these underlying variables that are prone to errors. For example, in neuroscience and genomics, the measured brain signals obtained by functional magnetic resonance imaging (fMRI) and the measured gene expression using RNA sequencing usually contain error through the measurement process [16, 22]. Another example is the responses to psychometric questionnaires where the questions represent noisy views of various traits [1; Scheines and Ramsey, 2016]. In such cases, it is important to use causal discovery methods capable of handling measurement errors, such as our proposed SEM-ME recovery algorithm. Additionally, a distinctive feature of our proposed SEM-ME, compared to previous work in the literature, is that we do not require all variables to have measurement errors. These variables can include, for example, age, race, gender, etc., which can be relevant in some of the aforementioned applications. That is, our model allows for applications containing a mixture of exactly measured variables as well as variables measured with error.
>
> Moreover, in most real-life setups we are not able to observe or measure many variables in the system (for instance in socioeconomical applications many variables are unobserved or even unknown) and hence, the violation of the causal sufficiency assumption is the norm rather than the exception. For example, in the field of non-invasive neuroimaging such as Electroencephalography (EEG), the activities of many of the brain regions are not recorded. Hence causal sufficiency cannot be assumed [Mastakouri et al, 2019]. Similarly, in stock market, returns may be confounded by several unmeasured economic and political factors [1]. In such cases it is important to use causal discovery methods capable of handling unobserved variables such as our proposed SEM-UR recovery algorithm.
>
> Therefore, we believe working with SEM-ME and SEM-UR that we considered in our work is an important step towards realistic models. We will add the discussion of the applications of our work above to the revised paper.
>
> We will upload revised supplementary material which includes the modifications listed above. These modifications appear in Appendix E, and will be implemented in the camera-ready draft if accepted.
>
> **References:**
>
> Wang Y S, Drton M. Causal discovery with unobserved confounding and non-Gaussian data[J]. arXiv preprint arXiv:2007.11131, 2020.
>
> Chen W, Zhang K, Cai R, et al. FRITL: A Hybrid Method for Causal Discovery in the Presence of Latent Confounders[J]. arXiv preprint arXiv:2103.14238, 2021.
>
> Scheines R, Ramsey J. Measurement error and causal discovery[C]//CEUR workshop proceedings. NIH Public Access, 2016, 1792: 1.
>
> Mastakouri A, Schölkopf B, Janzing D. Selecting causal brain features with a single conditional independence test per feature[J]. Advances in Neural Information Processing Systems, 2019, 32.

---

### Official Review · Reviewer_WQ2G · 2022-07-11

**Rating:** 7
**Confidence:** 3
**Soundness:** 3 good
**Presentation:** 3 good
**Contribution:** 3 good

**Summary:**

The authors study causal discovery of linear models in the presence of measurement errors and unobserved confounders, for the situation where the mixing matrix can be identified up to column scaling and permutation. They show that there is a one-to-many relationship between models subject to measurement error and models with unobserved parentless causes. Given this mapping, they analyze the identifiability of both models and propose additional faithfulness assumptions for the two classes of models leading to identifiable structures, and in some cases, to identifiable parameters. Finally, they define graphical partitions (ordered groupings) that are invariant for the identified models.

**Questions:**

On page 7, line 300, why does the structure defined in Definition 4 lead to a star graph? This does not seem immediately obvious to me and perhaps should be explained in a few sentences. I would also briefly describe what a star graph is.

How many simulations were performed in Figure 4? If the experiments were replicated multiple times, as they should be, then why are there no confidence bars reported?

**Limitations:**

The authors have not discussed any potential negative societal impacts of their work, which can be excused for such a highly theoretical work. I concur with the authors' claim that causal structure learning from observational data is less prone to these impacts than randomized experiments.

**Strengths And Weaknesses:**

The analogy between linear causal models in the presence of measurement error and linear causal models with (top-level) latent variables is, to best of my knowledge, novel and constitutes an interesting perspective on causal discovery without causal sufficiency. The technical quality of the paper is high, the text is clearly written for the most part, and the results seem to bridge an important gap in the causal discovery literature. The simulation section is brief, but comprehensive enough.

On the negative side, confidence bars are missing from the simulation results in Figure 4, even though the authors mention in the checklist that they have been reported. The figure is a bit hard to read, and some minor writing errors hinder clarity.

Minor comments:
- p1 (page 1), l34 (line 34): "application" $\to$ 'applications'
- p1, l56: "casual" $\to$ 'causal'
- p3, l95: "latter" $\to$ 'later', to be consistent with p6, l289
- p6, l263: "passes" $\to$ 'pass'
- p7, l341: "exist" $\to$ 'exists'
- p8, l356: "parent" $\to$ 'parents'
- p8, l359: "child" $\to$ 'children'
- p8, l371: "the edge in" $\to$ 'edges in the'
- p8, l372: "are" $\to$ 'is'
- p8, l373: "group" $\to$ 'groups'
- p9, l409: add 'the' before "Reconstruction"

---

> ### Author Response · Authors · 2022-08-02
> **Response to Reviewer WQ2G**
>
> We thank the reviewer for the comments. We address the reviewer's concerns below.
>
> **Regarding the “Star graph” question:**
>
> Star graph means that there is a center node in the graph, and all other nodes (if exist) are only connected to the center node. Under the ancestral ordered grouping described in Definition 4, for a linear SEM-ME, the induced subgraph of the nodes belonging to each ancestral ordered group is a star graph (We emphasize that this is about the **induced** subgraph). This is because:
>
> 1. Each ancestral ordered group can include at most one `nu-leaf` node.
> 2. When a group has no `nu-leaf` nodes, then the group has only one `u-leaf` node. Otherwise, when a group has one `nu-leaf` node, then all other nodes in this group are `u-leaf` nodes that are children of the `nu-leaf` node.
> 3. There exist no connections among `u-leaf` nodes by definition.
>
> Therefore, the induced subgraph corresponding to each ancestral ordered group constitutes a star graph. Similar arguments hold for linear SEM-UR.
>
> For example, in Figure 2(a), the AOG of the model is {{$Z_1$}, {$Z_2$}, {$Z_3, Z_4$}} . The induced subgraphs on all three ancestral order groups ($Z_1$, $Z_2$, $Z_3 \rightarrow Z_4$) are star graphs.
>
> In the revised paper, we will add a footnote at the end of the sentence in line 300 explaining what a star graph is.
>
> **Regarding error bars and simulation results:**
>
> We repeated the simulation 50 times under each setting.
>
> The reason we did not report the error bars in Figure 4 in the main text is that the figure includes multiple algorithms/settings and the presentation becomes cluttered. Yet, we included the error bars of DOG recovery algorithm in Figure 10 on Page 25 (supplementary material). As mentioned above, under each setting of the experiment, we repeated the simulation for 50 randomly generated models and reported the average results over the random seed, i.e., the plots are not just for one specific underlying model. Therefore, the reported recovery performance is not based on a particular graph structure.
>
> Please refer to Appendix D for more details about the simulation results. Further, in the revised paper, we will:
>
> 1. Move the experiment details and explanation of the results from Appendix D to the main text.
> 2. Re-organize Figure 4 into 3 rows to increase the size of subfigures (to make it more readable per reviewer’s request). We will also add more details to the caption. For instance, lower SHD/Edge and higher F1-score indicate better performances.
> 3. Add two figures to the appendix in which, as we did in Figure 10, we will report the error bars for AOG recovery algorithm and ICA-LiNGAM. We will also re-organize Figure 10 to make it more readable.
>
>
>
>
> We will upload revised supplementary material which includes the modifications listed above. These modifications appear in Appendix E, and will be implemented in the camera-ready draft if accepted.

---

> > ### Comment · Reviewer_WQ2G · 2022-08-09
> > **Thanks**
> >
> > Thank you very much for the patient explanation regarding the star graph. Indeed, on my initial reading, I had missed that each induced subgraph is a star graph, and not the graph as a whole.
> >
> > Thank you for the clarification regarding the error bars, I'm happy with the changes you have proposed. I would perhaps also briefly mention the reason for not adding error bars to Figure 4 in the main text, with a reference to Appendix D.

---

> > > ### Author Response · Authors · 2022-08-09
> > > **Response**
> > >
> > > We thank the reviewer for the response. We will definitely implement these changes if the paper is accepted.

---

> ### Comment · Area_Chair_RvW8 · 2022-08-08
> **Response needed**
>
> Dear Reviewer WQ2G,
>
> Please kindly respond to the rebuttal provided by the authors and/or engage in the discussion with them. If it addresses your concerns, please react accordingly. Otherwise, please elaborate in your review on why you think the rebuttal/discussion is inadequate. Thank you.
>
> Best, AC

---

### Author Response · Authors · 2022-08-04
**Revised supplementary material uploaded**

We uploaded a revised version of the supplementary material where we added a section (Appendix E, page 26-29) to show the changes we will implement to our paper.

---

### Meta-Review · Area_Chair_RvW8 · 2022-08-28

**Recommendation:** Accept
**Confidence:** Less certain

**Metareview:**

There is a consensus among all expert reviewers that the paper provides solid and rigorous theoretical results that contribute to the sub-field of causal inference, especially the identifiability of linear non-Gaussian models **when there is a measurement noise**, although the authors could have done better at providing concrete motivation and demonstrating the real-world impact of the results. Some of the issues however have been addressed during the discussion phase. I do hope the authors will take reviewers' comments into consideration when preparing the camera-ready version of this paper.

The main contribution of this work is to show the analogy between linear causal models in the presence of measurement error and linear causal models **with parentless latent variables**. This constitutes an interesting perspective on causal discovery without causal sufficiency and enables the authors to obtain identifiability results based on the mixing matrix in both cases. The results, therefore, bridge an important gap in the causal discovery literature.

**Award:**

No

---

### Decision · Program_Chairs · 2022-09-14

Accept